# Model-based aviation advice on distal volcanic ash clouds by assimilating aircraft in-situ measurements

G. Fu[1], A.W. Heemink[1], S. Lu[1], A.J. Segers[2], K. Weber[3], and H.X. Lin[1]

[1]Delft University of Technology, Delft Institute of Applied Mathematics, Mekelweg 4, 2628 CD Delft, The Netherlands.
[2]TNO, Department of Climate, Air and Sustainability, P.O. Box 80015, 3508 TA Utrecht, The Netherlands.
[3]University of Applied Sciences, Environmental Measurement Techniques, Josef-Gockeln-Str. 9, 40474 Düsseldorf, Germany.

*Correspondence to:* G. Fu (G.Fu@tudelft.nl)

**Abstract.** The forecast accuracy of distal volcanic ash clouds is important for providing valid aviation advice during volcanic ash eruption. However, because the distal part of volcanic ash plume is far from the volcano, the influence of eruption information on this part becomes rather indirect and uncertain, resulting in inaccurate volcanic ash forecasts in these distal areas. In our approach, we use real-life aircraft in-situ observations, measured in the North-West part of Germany during the 2010 Eyjafjallajökull eruption, in an ensemble-based data assimilation system combined with a volcanic ash transport model to investigate the potential improvement on the forecast accuracy with regard to the distal volcanic ash plume. We show that the error of the analyzed volcanic ash state can be significantly reduced through assimilating real-life in-situ measurements. After a continuous assimilation, it is shown that the aviation advice for Germany, the Netherlands and Luxembourg can be significantly improved. We suggest that with suitable aircrafts measuring once per day across the distal volcanic ash plume, the description and prediction of volcanic ash clouds in these areas can be greatly improved.

## 1 Introduction

Ash produced during explosive volcanic eruptions can cause serious impacts close to the volcano as well as at great distances (Melville, 1986). Turbine engines are particularly threatened by ingestion of airborne ash, and aircraft surfaces may be subject to abrasion and in the longer-term corrosion (Casadevall, 1994). For example, the sudden eruption of the ice-capped Eyjafjallajökull volcano at 1666 m height in Iceland from 14 April to 23 May 2010, had caused an unprecedented closure of the European and North Atlantic airspace resulting in a huge global economic loss (Bonadonna et al., 2012). Due to the huge impacts on aviation community, a lot of research has been initiated on how to efficiently reduce these aviation impacts starting with improving the accuracy of volcanic ash forecast after eruption onset (Gudmundsson et al., 2012; Eliasson et al., 2011; Schumann et al., 2011). Currently a lot of approaches, employing satellite-based (Prata and Prata, 2012; Stohl et al., 2011; Lu et al., 2016) or ground-based (Emeis et al., 2011) measurements, focus on improving the estimation of Eruption Source Parameters (ESPs) such as plume height and mass eruption rate. These are very important for a good estimation of volcanic ash emission. However, for the volcanic ash plume far from the volcano which could be very important for local aviation, more accurate ESPs alone will not be very useful. This is mainly because (1) compared to ESPs, the plume transport becomes more and more dominant as the distance to the volcano increases (Macedonio et al., 2016); (2) the small errors in ESPs can

accumulate into large errors in predicted ash concentrations after a large transport distance (Webster et al., 2012). Therefore, additional observation data, e.g., direct observations of distal volcanic ash plume must be employed to improve the aviation advice over continental Europe.

Ensemble-based data assimilation, which refers to the (quasi-) continuous use of the direct measurements to create accurate initial conditions for model runs (Zehner, 2010), is one of the most commonly used approaches for real-time forecasting problems (Evensen, 2009). In each assimilation step, a forecast from the previous model simulation is used as a first guess, then this forecast is modified to be more in agreement with the available observations. This approach is very effective for regional forecasting. For employment of ensemble-based data assimilation, in-situ measurements are the optimal type of observations (Evensen, 2009). Although satellite measurements are considered as the most commonly used volcanic ash observations based on their large detection domain and long-time continuous output data, they are not directly suited for data assimilation systems. This is because satellite observations are often not direct measurements of the quantity of interest, but optical property measurements. Therefore the aerosol quantity needs to be derived by a retrieval process or a complex observation operator. Moreover, satellite data are often two-dimensional (2D), thus are lack of sufficient vertical resolution (Bocquet et al., 2015). Note that, some satellites can provide very detailed vertical information on plumes (e.g., Cloud-Aerosol Lidar with Orthogonal Polarization (CALIOP) lidar measurements) but are spatially sparse (Winker et al., 2012). Fortunately, the in-situ volcanic ash state variables can be directly and accurately measured nowadays by means of airborne observations of volcanic ash (Weber et al., 2012). These aircraft-based measurements can be obtained in the boundaries of volcanic ash plume, which are probably the most direct volcanic ash observations possible. They have some unique advantages. First, aircraft data is usually obtained from optical particle counters, thus real-time particle concentration observation can be directly measured (Weber et al., 2012). Second, the aircraft measurement is in-situ which can be compared directly to a 3-dimensional model state variable (e.g., concentration). Whereas other measurements such as satellite data and LIDAR data observe optical properties that contain indirect information about concentrations and require further modeling and assumptions for getting translated into concentration estimates. Note that in this study volcanic ash state refers to the whole volcanic ash plume, while volcanic ash state variables represent point-to-point volcanic ash concentrations inside the ash plume. Third, an aircraft is flexible in flight route to follow the ash clouds. Although an aircraft measurement plan is usually beforehand designed for a region/altitude of interest, the aircraft operator can adjust the detailed route according to the real conditions (e.g., ash concentration level, wind direction) of the plume transport in order to always obtain appropriate volcanic ash concentrations (Weber et al., 2012).

Recently in the application of volcanic ash transport, the benefit of aircraft in-situ observations in an Ensemble Kalman Filter (EnKF) system has been studied (Fu et al., 2015). It was shown using so-called twin experiments that ensemble-based data assimilation is in principle able to combine the aircraft in-situ measurements with a volcanic ash transport and dispersion model (VATDM) to make improvements on volcanic ash estimation close to the eruption location. In that study, the focus was on the near-volcano areas where the uncertainties on plume height and mass eruption rate turned out to have a large influence on the estimates of the forecasted ash concentrations. However, for distal volcanic ash plume, these eruption parameters hardly improve the forecasts over a long distance. A larger mass eruption rate may cause the distal volcanic ash plume to spread stronger and wider after a long time period. But this potential effect can be significantly influenced or even canceled out by

a combination of a number of elusive physical factors over a long time period such as wind speed and direction. Thus the results on near-volcano areas cannot be directly employed for far-volcano regions, e.g., central Europe in the case of a volcanic eruption in Iceland. In addition, the aircraft in-situ measurements used in the previous studies were self-designed (artificial) based on model simulations from which actual conditions might differ significantly. For example, using data of a period of 10 hours by an aircraft gives accurate assimilation results. But in practical situations, a continuous aircraft measurement mission is at most 3 or 4 hours, thus it is still uncertain whether the assimilation can produce significant effect with a shorter measurement mission. Therefore, in case of real-life aircraft in-situ measurements, it remains unknown whether the ensemble-based data assimilation still has significant improvements on the distal part of volcanic ash clouds and how long the influence will last. The answers of these questions will lead us to a solution for evaluating distal volcanic ash clouds and further provide accurate aviation advice. This study aims at investigating these questions. Note that, the term real-life aircraft measurements in this study refer to authentic measurements obtained by real aircrafts. This is to distinguish the artificial aircraft measurements as used in (Fu et al., 2015). Another term distal volcanic ash plume is used to clarify the study focuses on volcanic ash forecasts far from the volcano, i.e. continental Europe in this study.

This paper is organized as follows. Section 2 introduces the LOTOS-EUROS model, aircraft in-situ measurements, and ensemble-based data assimilation methods used in this study. The assimilation experiments on distal volcanic ash clouds are specified in Section 3. Section 4 validates the performance of real-life data assimilation. Section 5 contains the benefit of the improved forecasts of distal ash plume on aviation advice, and also how much and how long the benefit has effect. Finally, the last section summarizes the concluding remarks of our research.

## 2    Materials and methods for volcanic ash assimilation

### 2.1    The LOTOS-EUROS model for volcanic ash transport

In this study, we use the LOTOS-EUROS model (Schaap et al., 2008) to simulate volcanic ash transport and dispersion, which is an operational air-quality model, used for daily air quality forecasts over Europe (Curier et al., 2012), focusing on ozone, nitrogen oxides, and particular matter. The model uses the off-line meteorological data produced by European Center for Medium-Range Weather Forecasts (ECMWF). The model is used to produce volcanic ash dispersion simulations in a timely and useful manner for forecasting. To describe a volcanic eruption in LOTOS-EUROS model, Eruption Source Parameters (ESP) such as plume height (PH), mass eruption rate (MER), vertical mass distribution (VMD) and particle size distribution (PSD) are needed. Typically ESPs for different volcanoes are provided as a look up table (Mastin et al., 2009).

The input parameter PH in LOTOS-EUROS is taken from the hourly based plume height detection by Icelandic Meteorological Office (IMO) (Gudmundsson et al., 2012) and usually the uncertainty of PH is taken as 20 % (Bonadonna and Costa, 2013). For VMD, large explosive volcanic plumes have a typical 'umbrella' shaped vertical distribution (Sparks et al., 1997) and as such this 'umbrella' shaped VMD is adapted into LOTOS-EUROS (Fu et al., 2015; Lu et al., 2016). The PSD in LOTOS-EUROS is defined as the ESP type S2 (see definitions in Mastin et al. (2009)), in which the mass fraction of erupted debris finer than 63 $\mu m$ is 0.4. For the S2 type eruption, Durant and Rose (2009) provided the base for the PSD from their

analysis of the 1992 Crater Peak, Mount Spurr event. Hence, Table 1 shows the ash distribution based on their analysis and is used in our experiments. Another input parameter MER is very hard to measure for an explosive onsetting volcano. Usually it is calculated from the plume height. Mastin et al. (2009) studied the relation between these parameters and concluded that an empirical relationship between plume height PH(km) and mass eruption rate MER (kg s$^{-1}$) is

$$5 \quad \text{PH} = 2.00\text{V}^{0.241}, \quad and \quad \frac{\text{V}}{\text{MER}} = \frac{1.5e^3}{4.0e^6}. \quad (1)$$

where V(m$^3$ s$^{-1}$) represents the volumetric flow rate.

Recently, the LOTOS-EUROS model has been evaluated as an appropriate volcanic ash transport model (Fu et al., 2015) (Fig. 1**a**), where the volcanic ash concentrations are described by 6 aerosol tracers including PM$_{10}$ and PM$_{2.5}$ (corresponding to vash_5 and vash_6 in Table 1) to model the transport process. The physical processes that are relevant for volcanic ash are 10 similar as those that apply for mineral dust, e.g., advective transport and diffusion, deposition, coagulation, sedimentation, and resuspension (Langmann, 2013). Where the transport is determined by the wind fields that could be regarded as rather well known, the other processes deposition and sedimentation processes are rather uncertain. The parameterizations for the later processes involve assumptions on the particle distribution for example, which is difficult to summarize in a few numbers of bins. These processes act on the distribution of the total ash mass over the modes (particle sizes) and the total mass load; one 15 could therefore state that almost everything in the description of an ash cloud is uncertain, except for the cloud shape and position. The processes included in this study are transport, sedimentation, and wet- and dry-deposition, where the relevant properties such as average particle size (Table 1) are implemented following Zhang (2001). Processes that are missing yet are for example coagulation, evaporation, and resuspension, which might be considered in future when appropriate observations are available to constrain them, for example sedimentation amounts.

20 **2.2 Real-life aircraft in-situ measurements**

During the period of the Eyjafjallajökull eruption in April – May, 2010, the outskirts of the eruption plume were entered directly by research flights, delivering most direct measurements within the eruption plume during this eruptive event (Weber et al., 2012). The measurement aircraft (Fig. 1**b**) was equipped with optical particle counters (OPC) for in-situ measurements. Real-time monitoring of the particle concentrations was possible during the flights and in-situ measurements from the eruption 25 plume were obtained with high time- and spatial-resolution. Through a direct laboratory calibration experiment, in which the mass concentration obtained with the OPC was compared with the absolute mass concentration gathered on a gravimetric filter, the standard deviation between the gravimetric measurement and the OPC was estimated at 10% (Weber et al., 2010) which can be taken as the instrumental error for this type of measurements during the performed flight. The total measurement error not only contains the instrumental error, but also includes an estimate of the model representation error (Fu et al., 2015). 30 In this study we used the aircraft-based measurements taken by one measurement flight on 18 May, 2010 performed by the group Environmental Measurement Techniques at Duesseldorf university of Applied Sciences. The measurements took place at heights around 3 km (Weber et al., 2012) and in the North-West part of Germany including the border between the Netherlands and Germany, see Fig. 1**b** (the black rectangular area in Fig. 1**a**). The aircraft took off from the airfield "Schwarze Heide" in

the Northern part of the Rhein-Ruhr area, headed along the Dutch border in the direction of the North Sea, continued towards Hamburg and then returned to the airfield. Along the route, concentrations of $PM_{10}$ and $PM_{2.5}$ were measured, see Fig. 1**c** and 1**d**.

## 2.3  The Ensemble Kalman Filter

The ensemble-based data assimilation technique used in this study is an Ensemble Kalman Filter technique (EnKF). Apart from the original formulation (Evensen, 1994), other formulations have been introduced such as the Ensemble Kalman Smoother (EnKS) (Evensen and van Leeuwen, 2000), Ensemble Square Root Filter (EnSR) (Evensen, 2004), Reduced Rank Square Root Filter (RRSQRT) (Verlaan and Heemink, 1997), etc. Ensemble-based assimilation is easy to implement, suitable for real-time estimation of concentrations and has a very general statistical formulation.

The Ensemble Kalman Filter essentially is a Monte Carlo ensemble-based method (Evensen, 2003), based on the representation of the probability density of the state estimate in an ensemble of $N$ states, $\xi_1, \xi_2, \cdots, \xi_N$. Each ensemble member is assumed to be a single sample out of a distribution of the true state. The number of required ensemble members depends on the complexity of the probability density function to be captured, which is usually determined by the nonlinearity of the model and the description of the involved uncertainties. For volcanic ash assimilation, an ensemble size of 50 is considered acceptable in

terms of accuracy while keeping computation time within reach (Fu et al., 2015). For application of the filter algorithm to the LOTOS-EUROS model, in the first step of this algorithm an ensemble of $N$ volcanic ash states $\xi^a(0)$ is generated to represent the uncertainty in the initial condition $\mathbf{x}(0)$. In the second step, the forecast step, the LOTOS-EUROS model (with stochastic plume height) propagates the ensemble members from the time $k-1$ to $k$:

$$\xi_j^f(k) = M_{k-1}(\xi_j^a(k-1)). \tag{2}$$

The state-space operator $M_{k-1}$ describes the time evolution from the time $k-1$ to $k$ of the state vector which contains the ash concentrations in the model grid boxes. The filter state is a stochastic distribution with mean $\mathbf{x}^f$ and covariance $\mathbf{P}^f$ following:

$$\mathbf{x}^f(k) = [\sum_{j=1}^{N} \xi_j^f(k)]/N, \tag{3}$$

$$\mathbf{L}^f(k) = [\xi_1^f(k) - \mathbf{x}^f(k), \cdots, \xi_q^f(k) - \mathbf{x}^f(k)], \tag{4}$$

$$\mathbf{P}^f(k) = [\mathbf{L}^f(k)\mathbf{L}^f(k)^T]/(N-1), \tag{5}$$

where the superscript "$T$" represents the transpose of the matrix. The observational network is defined by the observation operator $H$ that maps state vector $\mathbf{x}$ to observation space $\mathbf{y}$:

$$\mathbf{y}(k) = H_k(\mathbf{x}(k)) + \mathbf{v}(k), \quad \mathbf{v}(k) \sim N(0, \mathbf{R}), \tag{6}$$

where the observation error $\mathbf{v}$ is drawn from a Gaussian distribution with zero mean and covariance matrix $\mathbf{R}$. Here, $\mathbf{y}$ contains aircraft in-situ measurements of ash concentration and $\mathbf{R}$ is filled in a diagonal matrix with the square of the standard deviation

of $y$. The operator $H$ then selects the grid cell in $x$ that corresponds to the observation location. When measurements become available, the ensemble members are updated in the analysis step using the Kalman gain:

$$\mathbf{K}(k) = (\mathbf{f} \circ \mathbf{P}^f(k))\mathbf{H}(k)^T[\mathbf{H}(k)(\mathbf{f} \circ \mathbf{P}^f(k))\mathbf{H}(k)^T + \mathbf{R}]^{-1}, \tag{7}$$

$$\xi_j^a(k) = \xi_j^f(k) + \mathbf{K}(k)[\mathbf{y}(k) - \mathbf{H}(k)\xi_j^f(k) + \mathbf{v}_j(k)], \tag{8}$$

where $\mathbf{v}_j$ represents realizations of the observation error $v$.

In Eq. (7), a distance-based localization (Houtekamer and Mitchell, 1998, 2001) is obtained using a Schur product $\mathbf{f} \circ \mathbf{P}^f$ (i.e., element-wise multiplication) in order to reduce the spurious correlations caused by the finite ensemble size, which is a general problem in ensemble-based data assimilation. The correlation matrix $\mathbf{f}$ is obtained by applying a correlation function to the Euclidean distance between two points. The correlations decrease to zero beyond a certain distance. Distance-based localization

can be easily implemented for eliminating the spurious correlations outside the localized region. For some applications (e.g., ozone, $CO_2$, sulfur dioxide), this approach has achieved an acceptable performance with a simple setup using a constant localization parameter of 50–100 km (Curier et al., 2012; Chatterjee et al., 2012; Barbu et al., 2009). In this study, 100 km is adopted as the localization parameter in volcanic ash assimilation of aircraft in-situ measurements.

## 3    Sequentially assimilating real-life aircraft in-situ measurements for distal volcanic ash clouds

### 3.1    Experimental Setup

As described in Section 2, an Ensemble Kalman Filter (EnKF) is used in this study to assimilate real-life aircraft in-situ observations. The LOTOS-EUROS model run starts at 09:00 UTC 14 April 2010 by considering a zero initial condition, equivalent to an assumption of "no ash load yet". The volcanic ash is released during the first guess forecast based on the defined ESPs (PH, MER, VMD, PSD), as discussed in Section 2.1. As the model state changes with the time in the numerical

simulation (the time step of model run is 10 minutes (Fu et al., 2015)), the model result from the previous time step is taken as the initial state for the next time step. When the model run is at 09:40 UTC 18 May, the volcanic ash state gets continuously modified by the data assimilation process through combining real-life aircraft-based measurements taken along the Dutch border until the time 11:10 UTC 18 May. The specification of uncertainties is essential for a successful data assimilation in this study. The stochastic PH is assumed to be temporally correlated and the correlation parameter $\tau$ is set to be 1 hour (Fu et al.,

2015). Thus, the PH noise ($N_{ph}$) at two times ($t_1$ and $t_2$) has the relation (Evensen, 2009) of $\mathbb{E}[N_{ph}(t_1) \cdot N_{ph}(t_2)] = e^{\frac{-|t_1 - t_2|}{\tau}}$, where $\mathbb{E}$ represents the mathematical expectation.

To assimilate measurements in a simulation model, it is necessary to quantify the model representation error. The model representation error is the discrepancy between the quantity that instrument observes, and what the model value represents. Concentration values are defined on discrete grids with a finite resolution at discrete time steps. A measurement location usually

does not coincide with the grid point where the concentration value is defined. The spatial resolution of the model used in this study is around 12 km $\times$ 12 km $\times$ 1 km, therefore the volume of one grid-box is about 150 km$^3$. Through model processing, the concentration of one grid-box represents an average value for this grid-box, while one aircraft in-situ measurement is a

point value in a 3 dimensional field. In this study, we choose the in-situ measurement corresponding to the grid-box average value. This approximation makes sense only when two assimilated measurements are positioned in two different grid-boxes. Considering the aircraft speed of 100-200 km h$^{-1}$ and the LOTOS-EUROS horizontal and vertical resolution, a 10 minutes assimilation frequency is chosen to guarantee different assimilated measurements are in different grid-boxes. The observation

therefore almost corresponds to one model state variable in a grid-box, which means the representation error of the model is probably small. For the moment we will therefore not explicitly specify a model representation error, but implicitly assume that it is zero. The total observation representation error, defined as the sum of the instrumental error and the model representation error, is taken as 10% in this study.

Since the real-life measurements of the PM$_{10}$ and PM$_{2.5}$ concentrations are available and the uncertainties of this type of

measurements are approximately known, ensemble-based data assimilation can be used to combine them with the LOTOS-EUROS model to reconstruct optimal estimates.

### 3.2    Evaluation of real-life data assimilation

It is first examined how the data assimilation actually performs in the system. Fig. 2 shows the measurements, the mean of the ensemble members, as well as the forecast and the analysis of selected ensemble members. From the estimation of both volcanic

ash components PM$_{10}$ and PM$_{2.5}$ (Fig. 2**a** and Fig. 2**b**), we find the forecast mean largely overestimates the measurement at every time step, but the overestimation diminishes by the assimilation process significantly. Instead, the analysis mean consistently approximates the measurements with a high accuracy. Note that, the high accuracy here doesn't mean "identically equal", but "very close". This result illustrates that the assimilation at the measurement location is able to approximate the observed values and also solves the problem of overestimation. Moreover, at the measurement location, the spread in the

analysis ensembles is much smaller than that in the forecast ensembles, which means the error variance of analysis value at the measurement locations is significantly reduced through the use of assimilation. This is because aircraft in-situ measurements are of high accuracy. Note that, Fig. 2**a** and Fig. 2**b** only show the ensemble mean and the spread of the ensembles at the measuring locations. However, we also want to know how much impact assimilation of aircraft in-situ measurements can have on a wide area of the distal volcanic ash plume. If the impact is only limited to the measurement locations or only a small

nearby area of ash plume, there will be no significant improvement in terms of aviation advice because flights need a rather large domain for safety guarantee.

In order to further investigate this effect, we first show the ensembles of uncertain volcanic ash simulations (see Fig. 2**c** and Fig. 2**d**), which correspond to the ash distribution at 11:10 UTC 18 May 2010 before and after assimilation. Note that, this study focuses on the distal volcanic ash plume, thus only the area of the whole plume marked as red rectangular as shown in

Fig. 1**a** is of interest. Without loss of generality, ensemble member 8 and 21 are chosen for illustrating the ensemble spread of the distal ash plume. Through comparing different ensemble members with respect to the forecast at 11:10 UTC (Fig. 2**c**), the ensemble forecast member 21 is shown to be very different from the member 8 in almost all the complete distal plume, thus the large error of the forecast is not only at measurement locations, but also in a large area around the measurement. Compared to the forecast, ensembles of analysis state (Fig. 2**d**) show no large differences across the entire domain of interest. This tells

us that assimilating aircraft measurements effectively reduce the ensemble spread of the whole distal ash plume, which is a sign of consistency to the measurement locations. Next, we investigate the assimilation impact on the ensemble mean over the distal volcanic ash plume. Fig. 3 shows examples of the mean at 09:40 UTC and 11:10 UTC 18 May with and without assimilating aircraft measurements. Compared to the case without assimilation at 11:10 UTC (Fig. 3**b**), large differences can
be observed in the simulation results with a continuous assimilation (Fig. 3**d**). Note that areas with ash concentration higher than 4000 $\mu$g m$^{-3}$ are classified as No Fly Zone (NFZ) (EASA, 2011; Fu et al., 2015), which means aviation in these areas is not allowed. After the assimilation process, the calculated volcanic ash concentrations in Germany, Luxembourg and the Netherlands (except in Northern Netherlands) have a lower concentration level (lower than 3000 $\mu$g m$^{-3}$) and the changes on volcanic ash state can be seen across a wide area. This is because the volcanic ash state variables become dependent and
correlated due to the transport process (advection and diffusion) and the temporal correlation of emission (Fu et al., 2015). Thus in a fairly large domain, the state change at measurement location also influences state variables in surrounding areas. This is caused by the chosen localization radius (see Section 2.3) in the assimilation process. Further, the downwind direction includes influenced state variables due to the transport of earlier corrected ash concentrations, especially regarding forecasts later than the assimilated time steps. Note that after a careful check on the wind field around the aircraft route, the term
"downwind" direction means the direction of "South-East", which will be used in the following discussions. Another note is that the differences between with and without assimilation are not obtained in one-time, but step by step with assimilating measurements over a period of one and a half hours from 09:40 UTC to 11:10 UTC. This can be seen from the assimilation results at 09:40 UTC and 11:10 UTC (Fig. 3**c** and Fig. 3**d**) where clear differences (Fig. 3**e** and Fig. 3**f**) between the two times can be observed and the effect of the assimilation at 09:40 UTC is less pronounced than at 11:10 UTC. This shows that after a
continuous assimilation of aircraft measurements, the differences with the original simulation are the result of an accumulation of all previous assimilation effects. This analysis also tells us that all the assimilation steps are important for the final result and that only using one or two measurements does not produce accurate results.

## 4   Validation of assimilation performance

Based on the analysis above, significant differences between volcanic ash simulations without and with assimilation have been
revealed. To examine whether the assimilated results are indeed more accurate than the model results, a further validation must be conducted. Fig. 4**a** and Fig. 4**b** show the comparison of the forecasted volcanic ash plumes with and without assimilation. The basic idea of this validation is to compare future in-situ measurements with the forecast of the volcanic ash plumes initiated with Fig. 3**b** and Fig. 3**d**. After the assimilation process, the assimilation influenced region temporally propagates to the downwind direction due to the meteorological drive (wind speed and direction). Thus the forecasted downwind ash
concentrations are influenced along the length of integration time after assimilation. Because all the other settings in the system are the same, a better forecast is expected due to a more accurate initial state. We use the measurements from 09:30 UTC to 11:10 UTC along the Dutch border to produce the assimilated results, then we validate the results using another set of

aircraft in-situ measurements in the downwind direction taken from 14:10 UTC to 15:00 UTC 18 May 2010 (see Fig. 1**b** and 1**d**). The validation data is selected carefully with respect to the influenced area (Fig. 3**f**).

With different initialization, the forecast of volcanic ash concentration at 15:00 UTC shows large differences. The forecast after assimilation (see Fig. 4**b**, lower than 3000 $\mu$g m$^{-3}$ in the downwind direction of the measurement track) is much smaller than that without assimilation (Fig. 4**a**, higher than 4000 $\mu$g m$^{-3}$ in some areas of continental Europe). Note that the forecast for both cases may be performed better by combining adjustments to state variables as well as eruption parameters. The detailed ash concentrations of two forecasts are compared with measurements in Fig. 4**c** and 4**d**. Both forecasts are shown to overestimate the measurements. This is in accordance with practical experience that volcanic ash simulations often overestimate the truth to guarantee a safe aviation advice. This is because in practice, until carefully designed engine performance tests are conducted in realistic volcanic ash cloud conditions, a cautious approach (overestimation) to advising commercial jet operations in airspace affected by volcanic ash is recommended (Prata and Prata, 2012). Furthermore, we can also see that at each validation location, the forecast with assimilation is closer to the measurements than the forecast without assimilation, and also that the overestimation is significantly reduced using assimilation. This shows that the forecast at these locations with assimilation is more accurate than the forecast without assimilation, therefore the assimilated volcanic ash state (Fig. 3**d**) is a more accurate approximation to the real state of distal volcanic ash plume. In addition, we conclude that the assimilation process performs well in combining with the LOTOS-EUROS transport model with real-life measurements.

## 5 Assimilation benefit for aviation community

Next it will be investigated what is the benefit of the improved forecasts of distal ash plume on aviation advice, and also how much and how long the benefit has an effect. Firstly, the assimilation impact in the downwind and upwind directions is considered. For this investigation, 5 big cities around the measurement route are selected (see Fig. 5**a**). They are Dortmund, Cologne, Luxembourg in the downwind direction and Amsterdam, Rotterdam in the upwind direction. The evaluation height is chosen at 3 km. For some continental or intercontinental passenger flights, 3 km might be of special interest regarding taking off and landing. The evaluation time is chosen to be 11:10 UTC 18 May 2010 when the assimilation process finishes. The concentrations of two major distal volcanic ash components, i.e., PM$_{10}$ and PM$_{2.5}$ (Webley et al., 2012; Fu et al., 2015), are evaluated. Fig. 5**b** shows that results with assimilation is lower for both PM$_{10}$ and PM$_{2.5}$ in all the selected cities. To quantify this impact on estimation of both ash components, an impact rate (IR) is introduced for quantification. The IR is defined as:

$$(\text{IR})_p(i) = \frac{(\text{SimuNoAssimi})_p(i) - (\text{SimuAssimi})_p(i)}{(\text{SimuNoAssimi})_p(i)}, \tag{9}$$

where $p$ means either PM$_{10}$ or PM$_{2.5}$, $i$ means index of selected cities. Moreover, $(\text{SimuNoAssimi})_p$ and $(\text{SimuAssimi})_p$ represent two simulations without or with assimilation. Using this equation, we can get the IR of all cities (see Fig. 5**b**). Based on the IR values, we find the assimilation impact in the downwind direction (Dortmund, Cologne and Luxembourg) are much more significant than those in the upwind direction (Amsterdam, Rotterdam). This means after assimilation, the most

significant impact on ash clouds is in the downwind direction where in this study it is mainly Germany (see assimilation impact areas in Fig. 3**f**).

The analysis above demonstrates that assimilating aircraft in-situ measurements has the ability to impact on regional volcanic ash clouds, especially in the downwind direction of the measurement route. It is also shown that assimilation has an impact on aviation advice. If there is no assimilation employed (see Fig. 3**b**), the volcanic ash concentration in the main transport direction of the distal ash plume reaches over 4000 $\mu$g m$^{-3}$. Thus, only relying on simulation results, the aviation advice on continental Europe is that the sky above the North Sea, the Netherlands and the western part of Germany is forbidden for flights. This aviation advice would shutdown flights in a large area. Because the Netherlands and Germany are important aviation hubs in Europe, imposing such a no-fly zone will affect all flights in the ash penetrated area and subsequently leads to a huge economic loss. In contrast, if based on the improved simulation after a continuous assimilation (Fig. 3**d**), the aviation advice would have been changed. The sky in large parts of Europe is open for commercial flights, because except in small parts of the Netherlands ash concentrations all over the domain of interest are lower than 3000 $\mu$g m$^{-3}$. This illustrates that the accuracy of aviation advice and the NFZ area can significantly benefit from the ensemble-based data assimilation process. Note that we give the aviation advice only on the strength of the results at 3 km height. Generally all model levels must be analyzed for real cases. And the real aviation advice also includes for which exact area and which time frame the advice is given.

Another question is how long the effect of improvement by assimilating aircraft measurements will last? The answer of this will provide us guidance on how often aircraft measuring should be performed. For investigating the time period of the assimilation impact, the volcanic ash plume is forecasted one day (Fig. 6) starting at 11:10 UTC 18 May 2010. Without loss of generality, PM$_{10}$ is chosen to analyze the forecast performance. 3 time snapshots in Fig. 6**a** – Fig. 6**c** are chosen to show the forecast differences between without and with assimilation. Since there are clear differences between the two cases, the assimilation impact can last one day. Note that this impact duration is only valid for the areas (especially for regions downwind to the assimilated observations), that are influenced by the assimilation, which changes with time. When forecasting 24 hours (Fig. 6**c**), differences still can be observed, but the impact of assimilation is obviously getting much smaller (compared to Fig. 6**a** and Fig. 6**b**). Small differences are visible at cloud boundaries. Actually we also examined the assimilation impact in the forecast of the next day and observed only very small differences. Therefore, the time period of the assimilation impact of this case study can be taken as 24 hours. From this analysis, we suggest the frequency of the measurement campaign to be once per day. This study can be used to provide guidelines for an optimal flight schedule in regional measurement tasks. Note that the impact time investigated is based on the meteorological information in distal volcanic ash plume during the period considered in this study. For other cases, the duration of effective assimilation could be differed.

## 6  Conclusions

In this study, aircraft in-situ measurements in distal volcanic ash clouds were assimilated in the LOTOS-EUROS model. During a continuous assimilation, the error of the analyzed volcanic ash state was significantly reduced through assimilating real-life in-situ measurements. The improved volcanic ash state after assimilation are the result of an accumulation of all

previous assimilation effects. It was shown that all the assimilation steps contribute to the final result. To examine whether the assimilated volcanic ash state were indeed more accurate than the conventional simulation, a validation with future in-situ measurements was conducted. The forecast with assimilation was shown more accurate than the conventional forecast without assimilation. It also concluded that the assimilation process performed well in combining with the LOTOS-EUROS transport

model with real-life measurements.

The validation results also revealed that with the transport models alone, it is difficult to accurately model volcanic ash movements. This is probably because model parameters (e.g., the plume height) are uncertain and some processes are missing, for example, coagulation, evaporation, and resuspension. Analysis of the results showed that data assimilation approach is able to compensate the model's deficiencies. Aircraft in-situ measurements have a high accuracy and plays an important role to a

successful data assimilation. The aircraft can enter the plume to selectively obtain observations, so that the measurements are in-situ and optimal for the ensemble-based data assimilation methodology.

Investigation was also carried out on the benefit of the improved forecasts of distal ash plume on aviation advice. We found that after assimilation, the most significant improvements on distal ash clouds are in the downwind direction where in this study it is mainly Germany. This phenomenon is due to the wind direction and the transport process during the continuous

assimilation. Investigation shows that the accuracy of aviation advice within the assimilation influenced area can significantly benefit from the ensemble-based data assimilation process. The computer experiment revealed that the time period of the improvement effect on the areas downwind to the assimilated observations can be taken as 24 hours. Based on this result, we suggest to schedule an aircraft measurement campaign at a frequency of once per day. This can be used to provide guidelines for planning future regional measurement tasks. The suggested frequency should be adjusted by the temporal strength (due to

wind induced transport) on the assimilation influenced area.

In this study, we applied an off-line approach for model running and simply used the deterministic meteorological input data. These data also contain uncertainties that influence ash cloud transport. In future work, in order to further improve the accuracy of ash forecasting, uncertainties in the meteorological data such as wind speed should also be considered. In this study, only aircraft in-situ measurements are used in a data assimilation system. We may expect that with other types of measurements

(e.g., satellite-based or LIDAR-based) together, the assimilation results will be more practical since the aircraft measurements cannot be always obtained. However, for this multi-observation data assimilation, other problems need to be first considered such as insufficient vertical resolution in certain satellite data. This is a difficult aspect for assimilating these data in a three dimensional model, and will be investigated in our future work.

*Author contributions.* All authors participated in the design and analysis of the assimilation experiment. G.F., A.J.S. and S.L. carried out

the LOTOS-EUROS modeling volcanic ash transport. K.W. validated real-life aircraft in-situ measurements and provided them for the data assimilation experiment. G.F., H.X.L. and A.W.H. analyzed the results and wrote the paper with contributions from all co-authors.

*Acknowledgements.* In this paper, OpenDA software (www.openda.com) was used to perform ensemble-based data assimilation. We are very grateful to the editor and reviewers for their reviews and insightful comments. We thank Claire Taylor for her kindly assisting with correction

of English grammars. The authors also thank SURFsara, the Netherlands Supercomputing Centre in Amsterdam for providing the Cartesius cluster as the computing facility in the parallel computing experiments (contract No. SH-332-15).

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

**Table 1.** Volcanic ash particle size distribution and ash bins property for LOTOS-EUROS model simulation.

| Bins | Particle Diameter | Percent of Mass | Average Particle Size ($\mu m$) |
|------|-------------------|-----------------|----------------------------------|
| vash_1 | 250 to 2000 $\mu$m | 29 | 1125.00 |
| vash_2 | 63 to 250 $\mu$m | 31 | 156.50 |
| vash_3 | 30 to 63 $\mu$m | 12 | 46.50 |
| vash_4 | 10 to 30 $\mu$m | 18 | 20.00 |
| vash_5 | 2.5 to 10 $\mu$m | 8 | 6.25 |
| vash_6 | 0.0 to 2.5 $\mu$m | 2 | 1.25 |

Verlaan, M. and Heemink, A. W.: Tidal flow forecasting using reduced rank square root filters, Stochastic Hydrology and Hydraulics, 11, 349–368, doi:10.1007/bf02427924, 1997.

Weber, K., Vogel, A., Fischer, C., van Haren, G., and Pohl, T.: Airborne measurements of the Eyjafjallajökull volcanic ash plume over northwestern Germany with a light aircraft and an optical particle counter: first results, vol. 7832, pp. 78 320P–78 320P–15, doi:10.1117/12.869629, 2010.

Weber, K., Eliasson, J., Vogel, A., Fischer, C., Pohl, T., van Haren, G., Meier, M., Grobéty, B., and Dahmann, D.: Airborne in-situ investigations of the Eyjafjallajökull volcanic ash plume on Iceland and over north-western Germany with light aircrafts and optical particle counters, Atmospheric Environment, 48, 9–21, doi:10.1016/j.atmosenv.2011.10.030, 2012.

Webley, P. W., Steensen, T., Stuefer, M., Grell, G., Freitas, S., and Pavolonis, M.: Analyzing the Eyjafjallajökull 2010 eruption using satellite remote sensing, lidar and WRF-Chem dispersion and tracking model, Journal of Geophysical Research, 117, 2012.

Webster, H. N., Thomson, D. J., Johnson, B. T., Heard, I. P. C., Turnbull, K., Marenco, F., Kristiansen, N. I., Dorsey, J., Minikin, A., Weinzierl, B., Schumann, U., Sparks, R. S. J., Loughlin, S. C., Hort, M. C., Leadbetter, S. J., Devenish, B. J., Manning, A. J., Witham, C. S., Haywood, J. M., and Golding, B. W.: Operational prediction of ash concentrations in the distal volcanic cloud from the 2010 Eyjafjallajökull eruption, J. Geophys. Res., 117, D00U08+, doi:10.1029/2011jd016790, 2012.

Winker, D. M., Liu, Z., Omar, A., Tackett, J., and Fairlie, D.: CALIOP observations of the transport of ash from the Eyjafjallajökull volcano in April 2010, Journal of Geophysical Research: Atmospheres, 117, n/a, doi:10.1029/2011jd016499, 2012.

Zehner, C., ed.: Monitoring Volcanic Ash From Space, ESA communication Production Office, doi:10.5270/atmch-10-01, 2010.

Zhang, L.: A size-segregated particle dry deposition scheme for an atmospheric aerosol module, Atmospheric Environment, 35, 549–560, doi:10.1016/s1352-2310(00)00326-5, 2001.

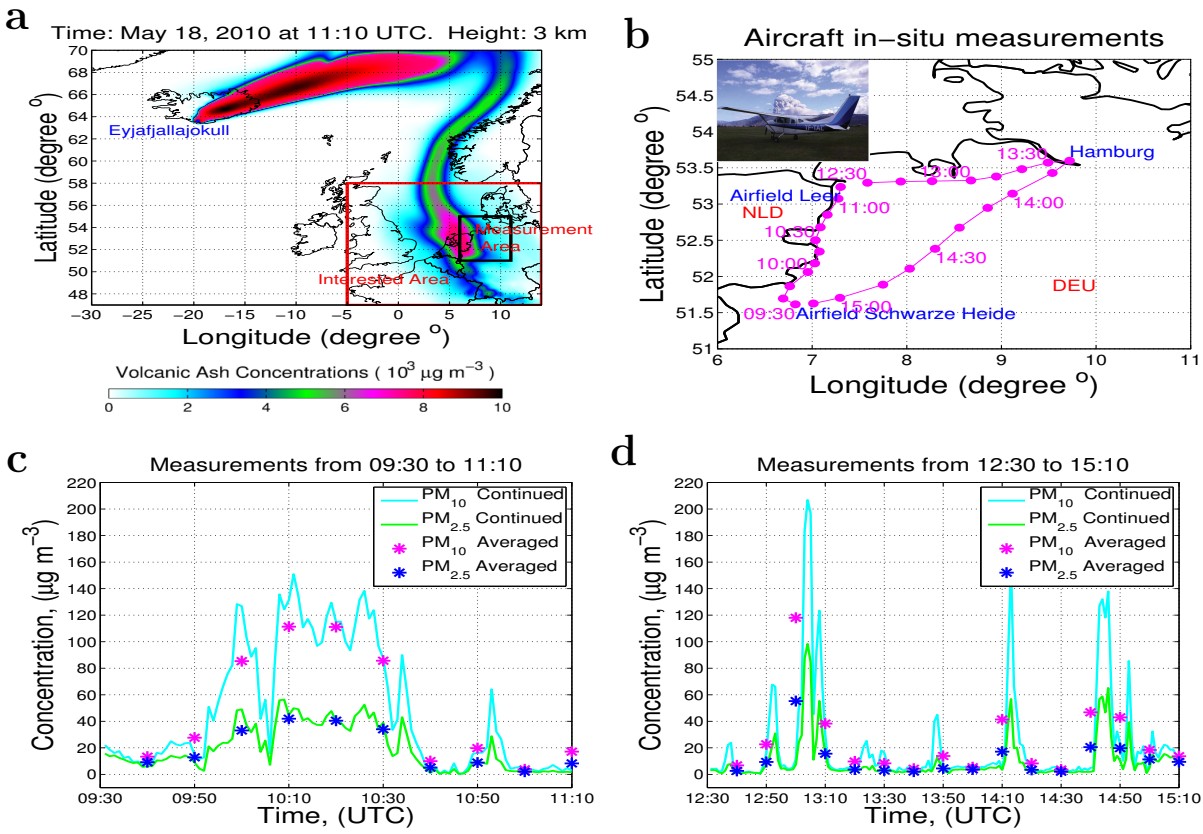

**Figure 1. Aircraft in-situ measurements of distal volcanic ash plume. a**, The LOTOS-EUROS simulation of volcanic ash plume at 11:10 UTC, 18 May 2010. **b**, Measuring aircraft flight route on 18 May 2010. **c**, $PM_{10}$ and $PM_{2.5}$ measurements from 09:30 UTC to 11:10 UTC. **d**, $PM_{10}$ and $PM_{2.5}$ measurements from 12:30 UTC to 15:10 UTC. In **c** and **d**, the curves show the values of $PM_{10}$ and $PM_{2.5}$ measured at a frequency of every 6 seconds. The values marked with a star are the averaged $PM_{10}$ and $PM_{2.5}$ (average every 10 minutes) which are used in the LOTOS-EUROS model in accordance with the model simulation step (10 minutes).

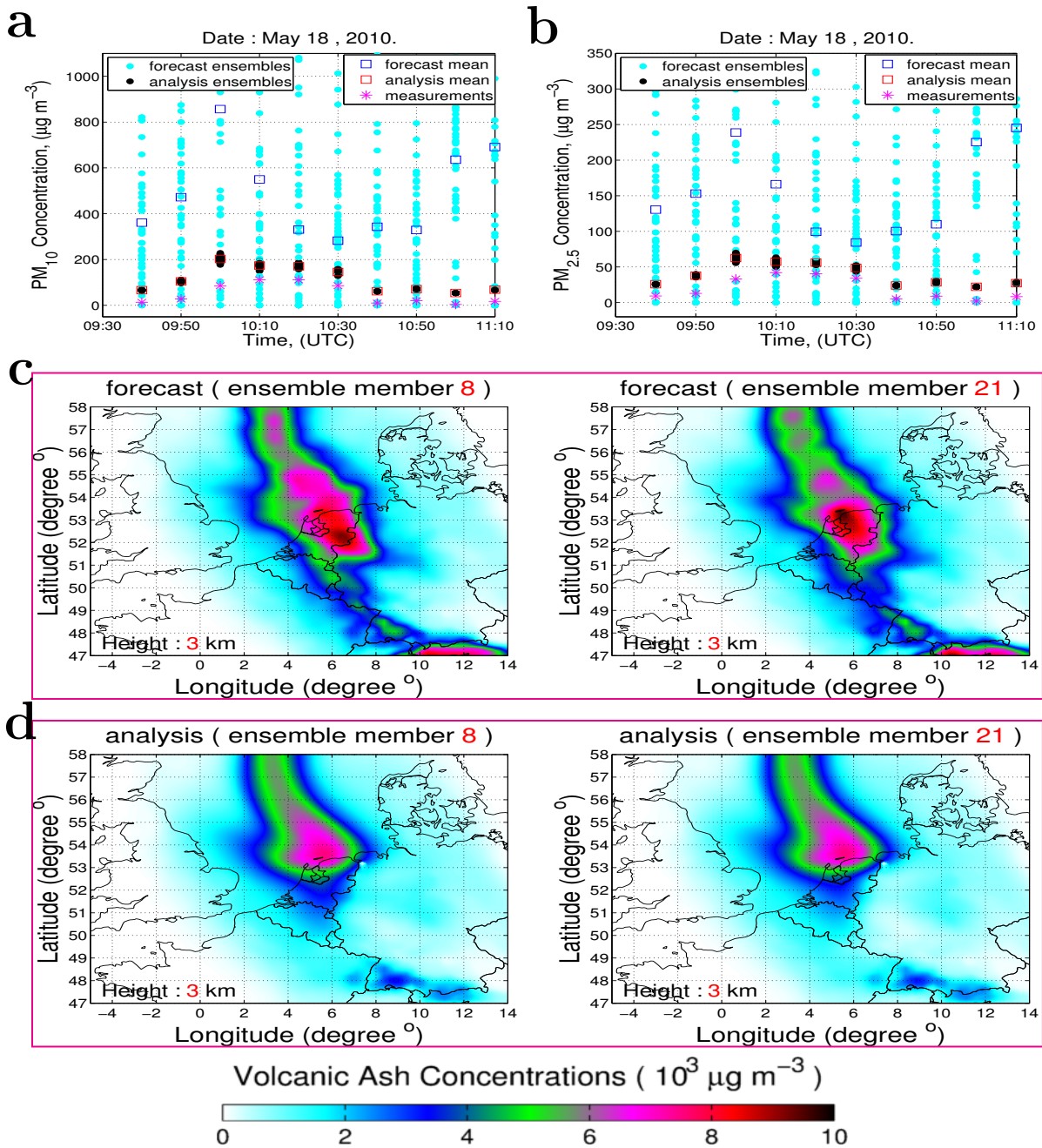

**Figure 2. Assimilation effect from 09:40 UTC to 11:10 UTC, 18 May, 2010. a** and **b**, Volcanic ash concentrations of $PM_{10}$ and $PM_{2.5}$ at measurement locations. **c** and **d**, Volcanic ash ensemble state (ensemble member 8 and 21) of forecast and analysis at 11:10 UTC. The measurements, ensembles and mean of forecast and analysis are shown in **a** and **b**. In **c** and **d**, the area of interest is marked as red rectangular in Fig. 1**a**. The evaluation height is chosen at 3 km since the measurements are taken at altitudes around this height, see Section 2.2.

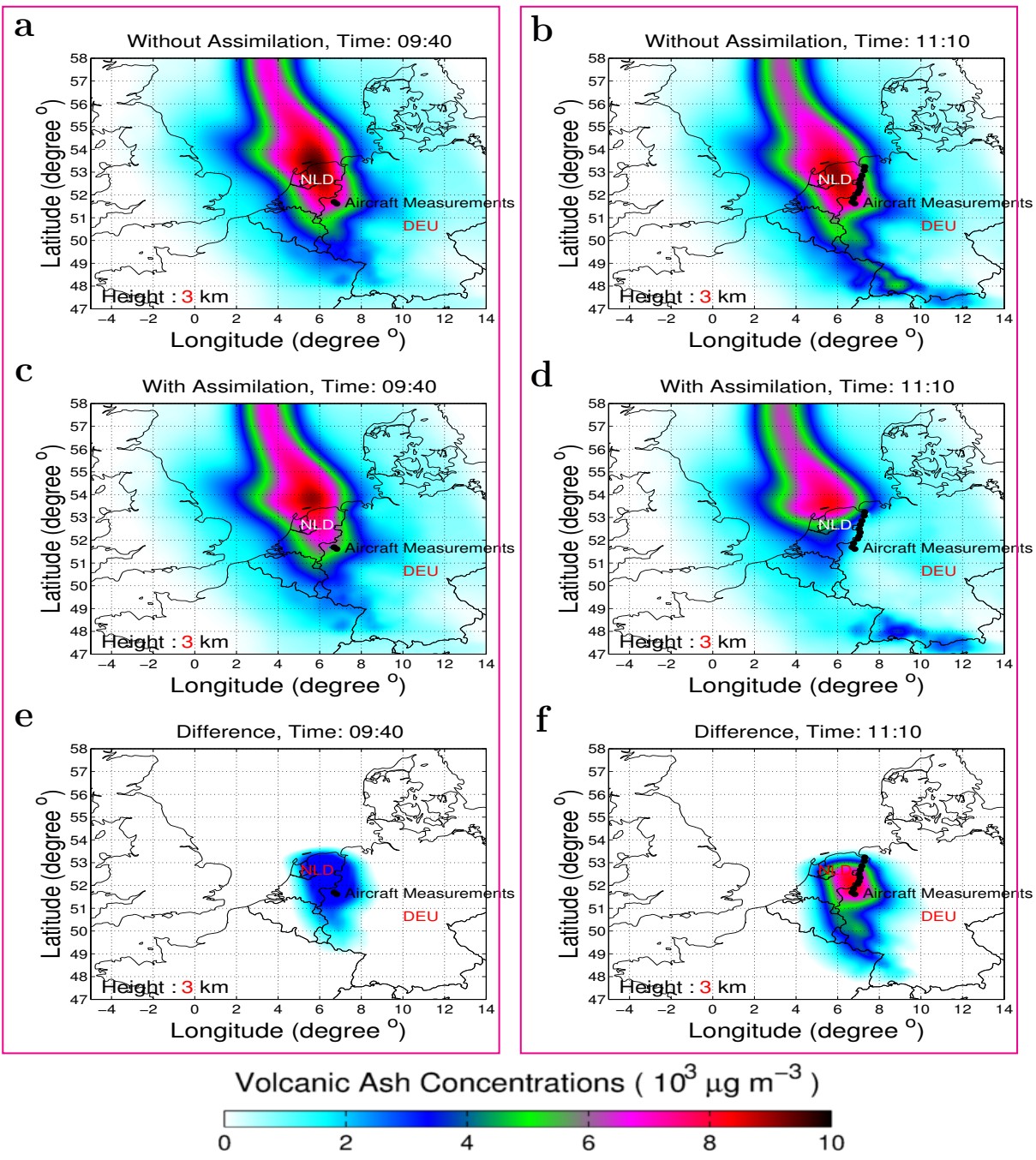

**Figure 3. Comparison with and without assimilating aircraft in-situ measurements on 18 May 2010. a** and **b**, Simulation results without assimilation at 09:40 UTC and 11:10 UTC. **c** and **d**, Simulation results with assimilation at 09:40 UTC and 11:10 UTC. **e**, Differences of **a** and **c**. **f**, Differences of **b** and **d**. The differences are in absolute values which are obtained by numerically subtracting the values between **a** and **c**, or **b** and **d**. **e** and **f** represent the areas where the assimilation has effect.

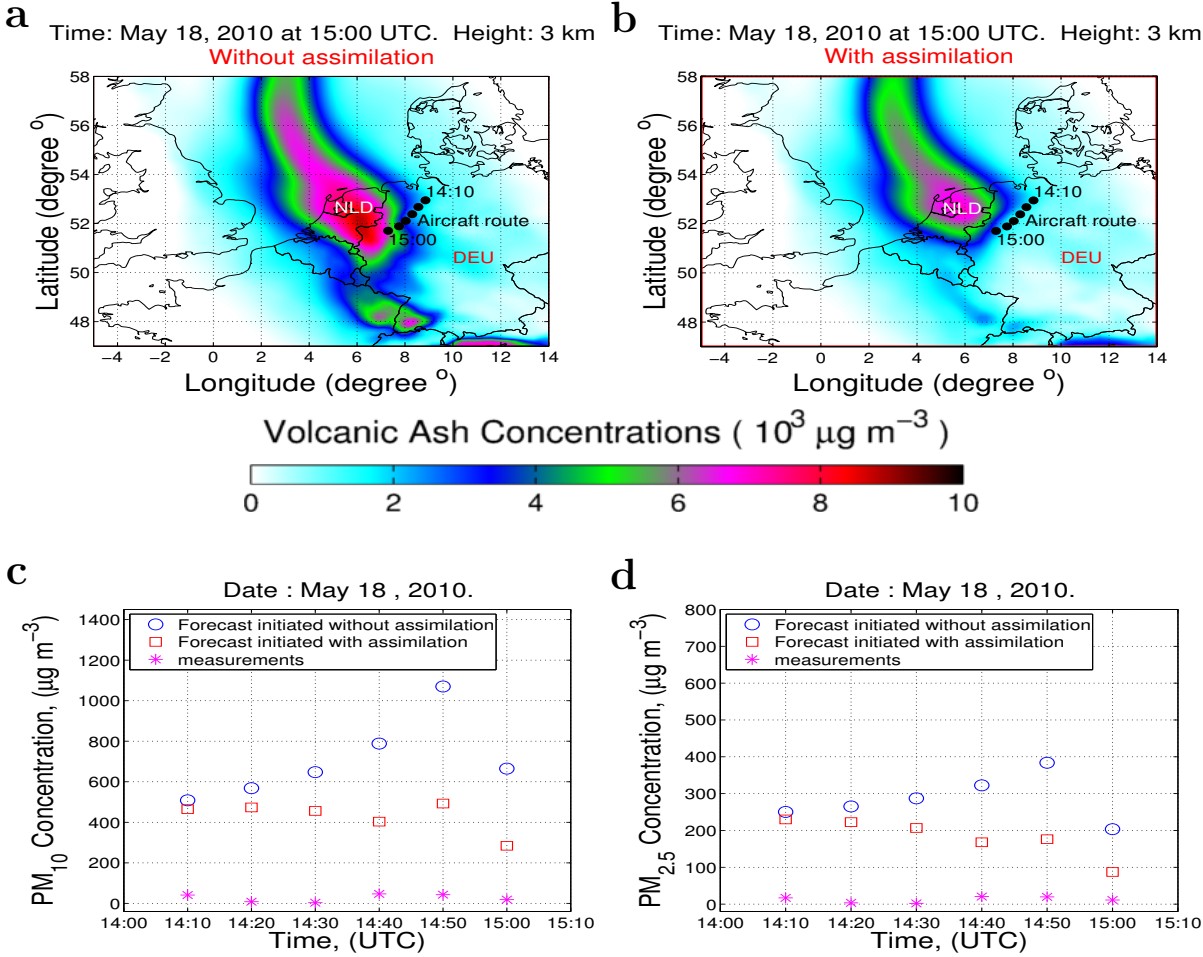

**Figure 4. Forecast at 15:00 UTC 18 May 2010 with different initial conditions for the volcanic ash state. a**, Forecast initiated with (Fig. 3**b**). **b**, Forecast initiated with (Fig. 3**d**). **c**, $PM_{10}$ concentration from 14:10 UTC to 15:00 UTC. **d**, $PM_{2.5}$ concentration from 14:10 UTC to 15:00 UTC.

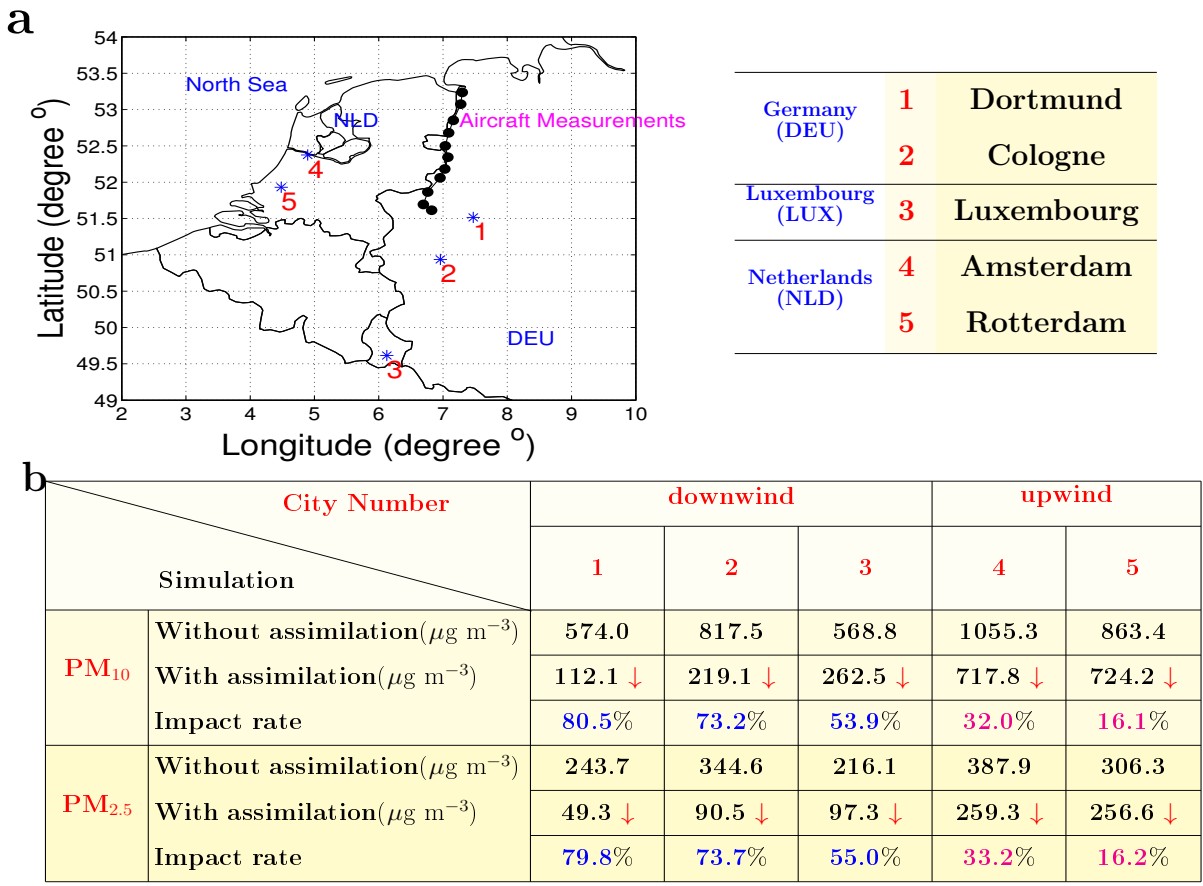

**Figure 5. PM$_{10}$ and PM$_{2.5}$ evaluation on selected cities with and without assimilation. a,** Selected international cities around the aircraft measurement track. City 1 – 3 are in the downwind direction, while city 4 – 5 are in the upwind direction. **b,** Concentrations of PM$_{10}$ and PM$_{2.5}$ and the quantified impact rates on selected cities. The height of interest is chosen at 3 km. The red arrow represents the trend of concentration values due to the assimilation process. The impact rates in downwind and upwind cases are distinguished by blue and magenta colors.

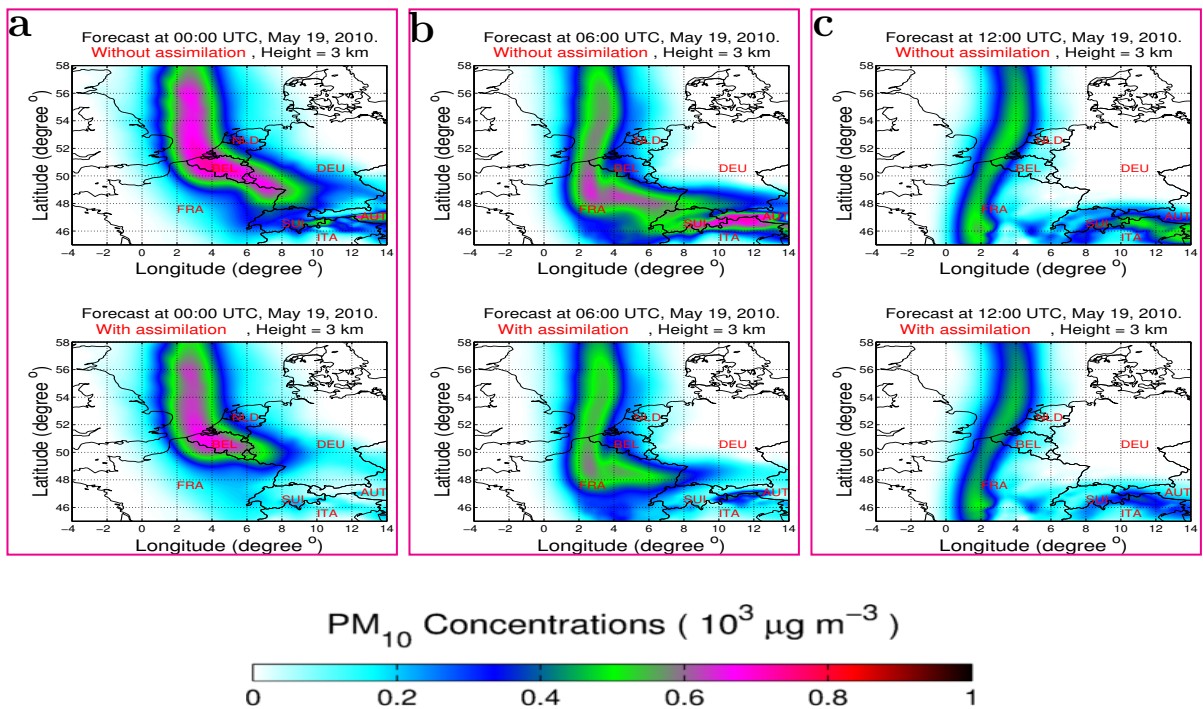

**Figure 6. One-day forecast of PM₁₀ concentration with and without assimilation.** A larger domain is chosen in this figure (compared to Fig. 2, 3, 4) to demonstrate the downwind change (with time) of the assimilation influenced area. **a**, Forecast at 00:00 UTC 19 May 2010. **b**, Forecast at 06:00 UTC 19 May 2010. **c**, Forecast at 12:00 UTC 19 May 2010.