# Peer review of "Model-based aviation advice on distal volcanic ash clouds by assimilating aircraft in-situ measurements"

_Atmospheric Chemistry and Physics, 2016_

## Referee Comment (RC1) · Anonymous Referee #1 · 22 Apr 2016

The subject of the article is to apply EnKF to a distal ash plume problem, using real observations. The authors include a cross validation that demonstrates improved forecasts within a short-range region of interest several hours after the observation time. This is an important work for devising improved aviation advice during volcanic eruptions, one that may inform future assimilation and aircraft campaigns. The paper however suffers from poor grammar (particularly towards the end), some extraneous figures, a thin discussion of model error and forecast improvement, and is also a bit thin in terms of background on the use of aircraft data for EnKF. While we are not requesting any further calculations, discussion of these issues and others as outlined below constitutes minor revisions prior to publication.

Comments: 2.10: This should be clarified to also acknowledge that some satellites provide very detailed vertical information on plumes (e.g., CALIOP) but are thus spatially sparse.

2.16: Actually, LIDAR gives gives profiles of e.g. aerosol extinction. I think the point the authors are trying to make here is that optical measurements are only an indirect constraint on concentration, which require further modeling and assumptions to translate into concentration estimates.

2.21: "These kinds of observations" is rather vague and potentially overly broad, as there have been many previous studies that have used EnKF to assimilate aircraft observations other than the authors' previous work. It seems that the background on the use of EnKF with satellite data for air quality forecasting could be expanded.

3.19: Splitting hairs on semantics, but this type of model can be evaluated but never really "validated", see Oreskes et al., Science, 1994.

3.22: It's not really clear how aerosols are treated based on this description. What does it mean "average particle size are implemented"? Perhaps a brief summary using customary descriptions of aerosol modeling theory would be helpful (internal or external mixture? fixed size, modal or sectional scheme? etc.).

Section 2.3. Add numbers to equations. I think the apostrophe is being used to represent transpose, which is a bit odd. Superscript T would be much more standard.

Page 4, Line 1; Page 5, Lines 2 and 21: Comment: The 10% measurement uncertainty is repeated in three places. Consider including it only in Section 2.2 or 3.1. At the very least, the first time this is mentioned it should be explicitly stated that it includes an estimate of the model error as well. But later, in section 3.1, perhaps it needs to be explained further how the value of the model transport error is quantified. Usually this takes some work, and properly identifying the horizontal and vertical correlation in the model error is rather critical tuning of a robust EnKF.

Figures 1b and 1c are unnecessary.

6.16 Bringing the analysis members into agreement in locations where there are no observations is only a sign of consistency, not accuracy. Independent measurements that are not assimilated are the only way to evaluate the forecasts away from the assimilation data.

7.26. This strikes me as a somewhat optimistic conclusion. It seems the evidence presented is that for some locations the forecast is improved, but in other it is not. Given the sparsity of the data used for assimilation, more often it is not. Rather, one might conclude that significantly more observations are required to completely constrain the ash plume across the domain. Extensive aircraft observations, or some combination of detailed aircraft measurements as used here with more broadly available remote sensing data may be useful.

7.10. An approach that combines adjustments to state variables as well as eruption parameters may perform better.

Section 5 and Figure 5. seem a bit flawed, in that they are presented in terms of an "Improvement rate", but all I see is a difference, without and with assimilation. Either a comparison to actual measurements needs to be included (or if that was actually the case, this needs to be made clear), or these need to be presented quite differently in terms of "impact rate", rather than "improvement rate".

6.16: "This tells us that assimilating aircraft measurements effectively reduce the error of the whole distal ash plume, not only at measurement locations." Comment: There could be some confusion in using the phrase "error of the whole distal ash plume", because there are not measurements to extract the model residual at all locations. Ensemble "spread" and "variance" are reduced.

9.17: "This is mainly because..." Comment: Do you know that "this is mainly because" or is it "probably because" or "possibly because"? There was no detailed process study

to ascertain all sources of error.

9.18: "Whereas…approach, the model's deficiencies can be compensated." Comment: Although they can be compensated, underlying deficiencies remain. The evidence lies in the failure to match cross validation data far away from assimilated measurements.

Corrections:

2.22: "It has been shown…" ->"It was shown…"

2.24: "In this study…" ->"In that study…"

6.20: "due to assimilation" ->"due to the assimilation"

8.18: "Note that areas with ash concentration higher than this value are classified as No Fly Zone (NFZ) …" Comment: It would be beneficial to mention this fact before Page 6, Line 21, where the concentration is mentioned. Give the context first.

7.2-3: Comment: This sentence needs revision for grammar.

7.24-26: Comment: This sentence needs revision for grammar.

7.30: Please correct to read as follows: "benefit has an effect. Firstly…impact in the downwind…directions is considered." Page 8, Line 2: "PM10 … evaluated." Comment: This sentence is grammatically confusing.

8.9: "we can find" -> "we find" 8.8-13: Comment: There are three sentences in a row about wind direction. Consider revising.

8.14: "measurements have the ability" -> "measurements has the ability" 8.15: "assimilation has impact" -> "assimilation has an impact"

8.17: "direction of distal"-> "direction of the distal"

8.20-23: "This aviation…forbidden area." Comment: This sentence has a lot of clauses and is confusing. Consider revising.

8.23-25: "In contrast…all lower than 3000ug/mˆ3." Comment: This sentence has a lot of clauses and is confusing. Consider revising.

8.31: "changes of the forecast differences between without and with assimilation" Comment: This phrase is very confusing. Consider revising.

8.31: "Since, there are" ->"Since there are"

8.34: "detected" ->"visible" or "noticeable" Comment: "detected" implies a physical observation.

8.34: "Actually we also examined the assimilation impact in the forecast of the next day, the difference was shown to be very small." Comment: This is a run-on sentence. Either add a conjunction or appropriate punctuation.

9.7-15: Combine these two paragraphs as they both relate to assimilation accuracy

9.24: "This phenomenon is due to the wind direction and the transport process during the continuous assimilation." Comment: This line is copied verbatim from Page 8, Line 12.

9.26: Please correct to read as follows: "The computer experiment…" 9.26-28: "Based on this…regional measurement tasks." Comment: Clean this up a bit to distinguish whether "the aircraft campaign" is a specific plan or any in the future that might be planned.

9.30: "Actually these data also contain uncertainties which have an influence on ash cloud transport." ->"These data also contain uncertainties that influence ash cloud transport."

9.30-31: "In future work, … into account." Comment: This sentence needs editing for grammar.

---

## Author Comment (AC1) · 29 Apr 2016

Dear Anonymous Referee #1,

On behalf of all co-authors, first of all I would like to thank you for giving us very useful comments and suggestions. We really appreciate your detailed comments and suggestions, which certainly improve the quality of the paper. In the revision, we have carefully considered the suggested changes (grammar issues, extraneous figures, discussons of model error, forecast improvement and the background on the use of aircraft data for EnKF) and included them to the best of our ability.

In the following we will give our response to every comment. To make the changes easier to identify, we have numbered them.

Best regards,
G. Fu

(The revised manuscript is in the latter part of this pdf.)

**Reply to the comments and suggestions**:

1. *2.10: This should be clarified to also acknowledge that some satellites provide very detailed vertical information on plumes (e.g., CALIOP) but are thus spatially sparse.*

   Response: It is included in lines 2.10–2.12:
   "Note that, some satellites can provide very detailed vertical information on plumes (e.g., Cloud-Aerosol Lidar with Orthogonal Polarization (CALIOP) lidar measurements) but are thus spatially sparse (Winker et al., 2012)."

2. *2.16: Actually, LIDAR gives gives profiles of e.g. aerosol extinction. I think the point the authors are trying to make here is that optical measurements are only an indirect constraint on concentration, which require further modeling and assumptions to translate into concentration estimates.*

   Response: Yes, we agree. It is reformulated in lines 2.17–2.19:
   "Whereas other measurements such as satellite data and LIDAR data observe optical properties that contain indirect information about concentrations and require further modeling and assumptions to translate into concentration estimates. "

3. *2.21: "These kinds of observations" is rather vague and potentially overly broad, as there have been many previous studies that have used EnKF to assimilate aircraft observations other than the authors' previous work. It seems that the background on the use of EnKF with satellite data for air quality forecasting could be expanded.*

   Response: I agree that "These kinds of observations" is not clear here. "These kinds of observations" refers to aircraft ash measurements discussed in last paragraph. It is now changed to lines 2.22–2.23:
   "Recently in the application of volcanic ash transport, the benefit of aircraft in-situ observations in an Ensemble Kalman Filter (EnKF) system has been studied (Fu et al., 2015).

"

4. *3.19: Splitting hairs on semantics, but this type of model can be evaluated but never really "validated", see Oreskes et al., Science, 1994.*

   Response: It is corrected in line 4.1:
   " Recently, the LOTOS-EUROS model has been evaluated as an appropriate volcanic ash transport model (Fu et al., 2015)".

5. *3.22: It's not really clear how aerosols are treated based on this description. What does it mean "average particle size are implemented" Perhaps a brief summary using cus- tomary descriptions of aerosol modeling theory would be helpful (internal or external mixture? fixed size, modal or sectional scheme? etc.).*

   Response: I agree with you that it was not clearly described how aerosols are treated. In the revision, Table 1 is added to show the "average particle size" and customary descriptions of aerosol modeling are added in lines 3.21–4.11:

Table 1: Volcanic ash particle size distribution and ash bins property for LOTOS-EUROS model simulation.

| Bins | Particle Diameter | Percent of Mass | Average Particle Size ($\mu m$) |
|---|---|---|---|
| vash_1 | 250 to 2000 $\mu$m | 29 | 1125.00 |
| vash_2 | 63 to 250 $\mu$m | 31 | 156.50 |
| vash_3 | 30 to 63 $\mu$m | 12 | 46.50 |
| vash_4 | 10 to 30 $\mu$m | 18 | 20.00 |
| vash_5 | 2.5 to 10 $\mu$m | 8 | 6.25 |
| vash_6 | 0.0 to 2.5 $\mu$m | 2 | 1.25 |

"The input parameter PH in LOTOS-EUROS is taken from hourly based Icelandic Meteorological Office (IMO) plume height detection and usually the uncertainty of PH is taken as 20 % (Bonadonna and Costa, 2013). For VMD, large explosive volcanic plumes have a typical 'umbrella' shaped vertical distribution (Sparks et al., 1997) and as such this 'umbrella' shaped VMD is adapted into LOTOS-EUROS (Fu et al., 2015; Lu et al., 2016). The PSD in LOTOS-EUROS is defined in the ESP type S2 (see definations in Mastin et al. (2009)), in which the mass fraction of erupted debris finer than 63 $\mu m$ is 0.4. For the S2 type eruption, Durant and Rose (2009) provides the base for the PSD from their analysis of the 1992 Crater Peak, Mount Spurr event. Hence, Table 1 provides the ash distribution based on their analysis and is used by LOTOS-EUROS. Another input parameter MER is very hard to measure for an explosive onsetting volcano. Usually it is calculated from the plume height. Mastin et al. (2009) studied the relation between these parameters and concluded that an empirical relationship between plume height PH(km) and mass eruption rate MER (kg s$^{-1}$) is

$$PH = 2.00V^{0.241}, \quad and \quad \frac{V}{MER} = \frac{1.5e^3}{4.0e^6}. \tag{1}$$

where V(m$^3$ s$^{-1}$) represents the volumetric flow rate.

Recently, the LOTOS-EUROS model has been evaluated as an appropriate volcanic ash transport model (Fu et al., 2015) (Fig. 1**a**), where the volcanic ash concentrations are described by 6 aerosol tracers including $PM_{10}$ and $PM_{2.5}$ (corresponding to vash_5 and vash_6 in Table 1) to model the transport process. The physical processes that are relevant for volcanic ash are similar as those that apply for mineral dust, e.g., advective transport and diffusion, deposition, coagulation, sedimentation, and resuspension (Langmann, 2013). Where the transport is determined by the wind fields that could be regarded as rather well known, the other processes deposition and sedimentation processes are rather uncertain. The parameterizations for the later processes involve assumptions on the particle distribution for example, which is difficult to summarize in a few numbers of bins. These processes act on the distribution of the total ash mass over the modes (particle sizes) and the total mass load; one could therefore state that almost everything in the description of an ash cloud is uncertain, except for the cloud shape and position. The processes included in this study are transport, sedimentation, and wet- and dry-deposition, where the relevant properties such as average particle size (Table 1) are implemented following Zhang (2001). "

6. *Section 2.3. Add numbers to equations. I think the apostrophe is being used to represent transpose, which is a bit odd. Superscript T would be much more standard.*

   Response: The apostrophe was used in some book on data assimilation, I agree that Superscript "T" is much more commonly used for the transpose. Numbers have been added to all equations and the transpose is represented with Superscript T.

7. *Page 4, Line 1; Page 5, Lines 2 and 21: Comment: The 10% measurement uncertainty is repeated in three places. Consider including it only in Section 2.2 or 3.1. At the very least, the first time this is mentioned it should be explicitly stated that it includes an estimate of the model error as well. But later, in section 3.1, perhaps it needs to be explained further how the value of the model transport error is quantified. Usually this takes some work, and properly identifying the horizontal and vertical correlation in the model error is rather critical tuning of a robust EnKF.*

   Response: The 10% measurement uncertainty is now only included in Section 2.2 and 3.1.

   As recommended, in section 2.2 (lines 4.22–4.23), it is explictly stated:
   "The total measurement error not only contains the instrumental error, but also includes an estimate of the model representation error (Fu et al., 2015)."

   In section 3.1, we added a detailed explanation about the model representation error in lines 6.14–6.27:
   "To assimilate measurements in a simulation model, it is necessary to quantify the model representation error. The model representation error is the discrepancy between the quantity that instrument observes, and what the model value represents. Concentration values are defined on discrete grids with a finite resolution at discrete time steps. A measurement location usually does not coincide with the grid point where the concentration value is defined. The spatial resolution of the model used in this study is around 12 km $\times$ 12 km $\times$ 1 km, therefore the volume of one grid-box is about 150 $km^3$. Through model processing, the concentration of one grid-box represents an average value for this grid-box, while

one aircraft in-situ measurement is a point value in a 3 dimensional field. In this study, we choose the in-situ measurement corresponding to the grid-box average value. This approximation makes sense only when two assimilated measurements are positioned in two different grid-boxes. Considering the aircraft speed of 100-200 km h$^{-1}$ and the LOTOS-EUROS horizontal and vertical resolution, a 10 minutes assimilation frequency is chosen to guarantee different assimilated measurements are in different grid-boxes. The observation therefore almost corresponds to one model state variable in a grid-box, which means the representation error of the model is probably small. For the moment we will therefore not explicitly specify a model representation error, but implicitly assume that it is zero. The total observation representation error, defined as the sum of the instrumental error and the model representation error, is taken as 10% in this study. "

8. *Figures 1b and 1c are unnecessary.*

   Response: I agree with you. In the revision, Figures 1b and Figures 1c have been deleted.

[Figure]

Figure 1: **Aircraft in-situ measurements of distal volcanic ash plume. a**, The LOTOS-EUROS simulation of volcanic ash plume at 11:10 (UTC), 18 May 2010. **b**, Measuring aircraft flight route on 18 May 2010. **c**, PM$_{10}$ and PM$_{2.5}$ measurements from 09:30 to 11:10 (UTC). **d**, PM$_{10}$ and PM$_{2.5}$ measurements from 12:30 to 15:10 (UTC). In **c** and **d**, the curves show the values of PM$_{10}$ and PM$_{2.5}$ measured at a frequency of every 6 seconds. The values marked with a star are the averaged PM$_{10}$ and PM$_{2.5}$ (average every 10 minutes) which are used in the LOTOS-EUROS model in accordance with the model simulation step (10 minutes).

9. *6.16 Bringing the analysis members into agreement in locations where there are no observations is only a sign of consistency, not accuracy. Independent measurements that are not assimilated are the only way to evaluate the forecasts away from the assimilation data.*

   Response: It is indeed not correct to say the accuracy in locations where there are no observations. It has been changed in lines 7.19–7.21:
   "This tells us that assimilating aircraft measurements effectively reduce the ensemble spread of the whole distal ash plume, which is a sign of consistency to the measurement locations."

10. *7.26. This strikes me as a somewhat optimistic conclusion. It seems the evidence presented is that for some locations the forecast is improved, but in other it is not. Given the sparsity of the data used for assimilation, more often it is not. Rather, one might conclude that significantly more observations are required to completely constrain the ash plume across the domain. Extensive aircraft observations, or some combination of detailed aircraft measurements as used here with more broadly available remote sensing data may be useful.*

    Response: We agree that the conclusion was too optimistic. It is reformulated in lines 8.27–8.33:
    "This shows that the forecast at these locations with assimilation is more accurate than the forecast without assimilation, so that the assimilated volcanic ash state (Fig. 3d) is a more accurate approximation to the real state of distal volcanic ash plume. In addition, we conclude that the assimilation process performs well in combining with the LOTOS-EUROS transport model with real-life measurements. Note that we find that for some locations the forecast is improved, but in other it is not. Thus, significantly more observations are required to completely constrain the ash plume across the domain. Extensive aircraft observations, or some combination of aircraft measurements as used here with remote sensing data may be useful. "

11. *7.10. An approach that combines adjustments to state variables as well as eruption parameters may perform better.*

    Response: It has been added in lines 8.14–8.15:
    "Note that the forecast for both cases may be performed better by combining adjustments to state variables as well as eruption parameters."

12. *Section 5 and Figure 5. seem a bit flawed, in that they are presented in terms of an "Improvement rate", but all I see is a difference, without and with assimilation. Either a comparison to actual measurements needs to be included (or if that was actually the case, this needs to be made clear), or these need to be presented quite differently in terms of "impact rate", rather than "improvement rate".*

    Response: I agree that "impact rate" is the proper term to present the difference of the assimilation impact between downwind and upwind directions. As recommended, we changed Section 5 and Figure 5 in terms of "impact rate", see lines 9.3–9.18:
    "Firstly, the assimilation impact in the downwind and upwind directions is considered. For this investigation, eight big cities around the measurement route are selected (see Fig.

**5a**). They are Dortmund, Düsseldorf, Cologne in the downwind direction and Amsterdam, Rotterdam, Antwerp, Brussels in the upwind direction. The evaluation height is chosen at 3 km for relevant to the continental commercial aircraft safety. This is because most national and maybe some continental passenger flights are around this altitude, while intercontinental flights are at much higher altitude (Fu et al., 2015). The evaluation time is chosen to be 11:10 UTC 18 May 2010 when the assimilation process finishes. The concentrations of two major distal volcanic ash components, i.e., $PM_{10}$ and $PM_{2.5}$ (Webley et al., 2012; Fu et al., 2015), are evaluated. Fig. **5b** shows that results with assimilation is lower for both $PM_{10}$ and $PM_{2.5}$ in all the selected cities. To quantify this impact on estimation of both ash components, an impact rate (IR) is introduced for quantification. The IR is defined as:

$$(\text{IR})_p(i) = \frac{(\text{SimuNoAssimi})_p(i) - (\text{SimuAssimi})_p(i)}{(\text{SimuNoAssimi})_p(i)}, \tag{2}$$

where $p$ means either $PM_{10}$ or $PM_{2.5}$, $i$ means index of selected cities. Moreover, $(\text{SimuNoAssimi})_p$ and $(\text{SimuAssimi})_p$ represent two simulations without or with assimilation. Using this equation, we can get the IR of all cities (see Fig. **5b**). Based on the IR values, we find the assimilation impact in the downwind direction (Dortmund, Düsseldorf, Cologne and Luxembourg) are much more significant than those in the upwind direction (Amsterdam, Rotterdam, Antwerp and Brussels). This means after assimilation, the most significant impact on ash clouds is in the downwind direction where in this study it is mainly Germany (see assimilation impact areas in Fig. **3f**). "

13. *6.16: "This tells us that assimilating aircraft measurements effectively reduce the error of the whole distal ash plume, not only at measurement locations." Comment: There could be some confusion in using the phrase "error of the whole distal ash plume", because there are not measurements to extract the model residual at all locations. Ensemble "spread" and "variance" are reduced.*

    Response: As recommended, it is now changed to "spread", see lines 7.19–7.20:
    "This tells us that assimilating aircraft measurements effectively reduce the ensemble spread of the whole distal ash plume".

14. *9.17: This is mainly because. . ." Comment: Do you know that "this is mainly because" or is it "probably because" or "possibly because" There was no detailed process study to ascertain all sources of error.*

    Response: I agree that here it should be "probably because", see lines 10.19–10.20:
    "This is probably because model parameters (e.g., the plume height) are uncertain and some processes are missing, for example, coagulation, evaporation, and resuspension."

15. *9.18: "Whereas. . .approach, the model's deficiencies can be compensated." Comment: Although they can be compensated, underlying deficiencies remain. The evidence lies in the failure to match cross validation data far away from assimilated measurements.*

    Response: We agree and changed this in lines 10.20–10.22:
    "Analysis of the results showed that data assimilation approach is able to compensate the

model's deficiencies. However, the failure to match cross validation data far away from assimilated measurements indicate that although they can be compensated, the underlying deficiencies remain."

**Corrections**:

1. As suggested, "It has been shown. . ." is corrected to "It was shown" in lines 2.23-2.25:
   "It was shown using so-called twin experiments that ensemble-based data assimilation is in principle able to combine the aircraft in-situ measurements with a volcanic ash transport and dispersion model (VATDM) to make improvements on volcanic ash estimation near to the eruption. "

2. "In this study..." is corrected to "In that study" in lines 2.25–2.27:
   " In that study, the focus was on the near-volcano areas where the uncertainties on plume height and mass eruption rate turned out to have a large influence on the estimates. "

3. "due to assimilation" is changed to "After the assimilation" in lines 7.25–7.27:
   "After the assimilation process, the calculated volcanic ash concentrations in continental Europe have a lower concentration level (lower than 3000 $\mu$g m$^{-3}$) and the changes on volcanic ash state is shown in a wide area. "

4. *8.18: "Note that areas with ash concentration higher than this value are classified as No Fly Zone (NFZ) . . ." Comment: It would be beneficial to mention this fact before Page 6, Line 21, where the concentration is mentioned. Give the context first.*

   Now this is mentioned before the first concentration value is discussed, as in lines 7.24–7.27:
   " Note that areas with ash concentration higher than 4000 $\mu$g m$^{-3}$ are classified as No Fly Zone (NFZ) (EASA, 2011; Fu et al., 2015), which means aviation in these areas is not allowed. After the assimilation process, the calculated volcanic ash concentrations in continental Europe have a lower concentration level (lower than 3000 $\mu$g m$^{-3}$) and the changes on volcanic ash state is shown in a wide area."

5. The grammar problem in the sentence has been corrected, as in lines 8.4–8.5:
   "Based on the analysis above, significant differences between volcanic ash simulations without and with assimilation have been revealed".

6. The sentence is revised, see lines 8.27–8.29:
   "This shows that the forecast at these locations with assimilation is more accurate than the forecast without assimilation, so that the assimilated volcanic ash state (Fig. 3d) is a more accurate approximation to the real state of distal volcanic ash plume. ".

7. *7.30: Please correct to read as follows: "benefit has an effect. Firstly. . .impact in the downwind. . .directions is considered." Page 8, Line 2: "PM10 . . . evaluated." Comment: This sentence is grammatically confusing.*

   The description is modified in lines 9.2–9.4:
   "Next it will be investigated what is the benefit of the improved forecasts of distal ash plume on aviation advice, and also how much and how long the benefit has an effect. Firstly, the assimilation impact in the downwind and upwind directions is considered."

And in lines 9.8–9.10:
"The concentrations of two major distal volcanic ash components, i.e., $PM_{10}$ and $PM_{2.5}$ (Webley et al., 2012; Fu et al., 2015), are evaluated."

8. *8.9: "we can find" to "we find" 8.8-13: Comment: There are three sentences in a row about wind direction. Consider revising.*

As suggested, corrections are made in lines 9.14–9.18:
" Based on the IR values, we find the assimilation impact in the downwind direction (Dortmund, Düsseldorf, Cologne and Luxembourg) are much more significant than those in the upwind direction (Amsterdam, Rotterdam, Antwerp and Brussels). This means after assimilation, the most significant impact on ash clouds is in the downwind direction where in this study it is mainly Germany (see assimilation impact areas in Fig. 3f). "

9. "measurements have the ability" is corrected to "measurements has the ability" and "assimilation has impact" is corrected to "assimilation has an impact" in lines 9.19–9.21:
"The analysis above demonstrates that assimilating aircraft in-situ measurements has the ability to impact on regional volcanic ash clouds, especially in the downwind direction of the measurement route. It is also shown that assimilation has an impact on aviation advice."

10. "direction of distal" is corrected to "direction of the distal" in lines 9.21– 9.22:
" If there is no assimilation employed (see Fig. 3b), the volcanic ash concentration in the main transport direction of the distal ash plume reaches over 4000 $\mu$g m$^{-3}$. "

11. *8.20-23: "This aviation. . .forbidden area." Comment: This sentence has a lot of clauses and is confusing. Consider revising.*

As suggested, the sentence is revised as in lines 9.23–9.26:
"This aviation advice would shutdown flights in a large area. Because the Netherlands and Germany are important aviation hubs in Europe, imposing such a no-fly zone will prohibit flights entering continental Europe in the Eastern direction and subsequently lead to a huge economic loss."

12. *8.23-25: "In contrast. . .all lower than 3000$\mu$g m$^{-3}$." Comment: This sentence has a lot of clauses and is confusing. Consider revising.*

As suggested, the sentence is revised as in lines 9.26–9.28:
" In contrast, if based on the improved simulation after a continuous assimilation (Fig. 3d), the aviation advice would have been changed. The sky in almost the whole of Europe is open for commercial flights, because except in small parts of the Netherlands ash concentrations all over Europe are lower than 3000 $\mu$g m$^{-3}$. "

13. *8.31: "changes of the forecast differences between without and with assimilation" Comment: This phrase is very confusing. Consider revising.*

As suggested, the sentence is revised as in lines 9.33–9.34:
" 3 time snapshots in Fig. 6a – Fig. 6c are chosen to show the forecast differences between

without and with assimilation.   "

14. "Since, there are" is corrected to "Since there are" in line 9.34–10.1:
"Since there are clear differences between the two cases, the assimilation impact will last at least one day."

15. Thanks for pointing out the difference between "detected" and "visible" or "noticeable". It is corrected in lines 10.2–10.3:
"Only small differences are visible in the Northern part of Italy."

16. *8.34: "Actually we also examined the assimilation impact in the forecast of the next day, the difference was shown to be very small." Comment: This is a run-on sentence. Either add a conjunction or appropriate punctuation.*

    As suggested, the sentence is corrected in lines 10.3–10.4:
    "Actually we also examined the assimilation impact in the forecast of the next day and observed only very small differences. "

17. *9.7-15: Combine these two paragraphs as they both relate to assimilation accuracy.*

    The two paragraphs are now combined together, see lines 10.10–10.17.

18. *9.24: "This phenomenon is due to the wind direction and the transport process during the continuous assimilation." Comment: This line is copied verbatim from Page 8, Line 12.*

    The sentence is removed, it is now only mentioned in the conclusion (lines 10.27–10.28).

19. *9.26: Please correct to read as follows: "The computer experiment. . ." 9.26-28: "Based on this. . .regional measurement tasks." Comment: Clean this up a bit to distinguish whether "the aircraft campaign" is a specific plan or any in the future that might be planned.*

    The sentences are revised in lines 10.29–10.31:
    "The computer experiment revealed that the time period of the improvement effect can be taken as 24 hours. Based on this result, we suggest to schedule an aircraft measurement campaign at a frequency of once per day. This can be used to provide guidelines for planning future regional measurement tasks. "

20. "Actually these data also contain uncertainties which have an influence on ash cloud transport."  is corrected to "These data also contain uncertainties that influence ash cloud transport." in line 10.33.

21. *9.30-31: "In future work, . . . into account." Comment: This sentence needs editing for grammar.*

    It is corrected in lines 10.33–11.1:

[revised manuscript text omitted]

---

## Referee Comment (RC2) · Anonymous Referee #2 · 24 May 2016

General comments:

The article represents a nice study treating the application of an Ensemble Kalman Filter on the prediction of volcanic ash plumes distal to the emission source, exploiting the observational information of aircraft-performed mass concentration measurements. This is innovative in terms of volcanic ash dispersion forecast improvements for aviation advice because it exploits real observational in-situ data by assimilation. The authors discuss the results as well as validate the assimilation performance using independent measurements. However, especially the discussion of assimilation impacts on subsequent ash dispersion predictions and thereon based aviation advices suffer from imprecise definitions of influenced areas. Further, some stated terms, interpretations,

and conclusions (see below) are deceptive so that reformulation and clarification of these issues are requested, while additional calculations are not required. Furthermore grammatical corrections are necessary in some sentences.

Specific comments:

2.1: It should be clarified why the influence of ESPs weakens with the distance to the volcano.

2.9: The difficulty of satellite data assimilation should be discussed carefully. Satellite lidar instruments like CALIOP or CATS provide highly resolved aerosol profiles. The challenge of satellite data assimilation is that the observations are often no direct measurements of the quantity of interest, but optical property measurements. Therefore the aerosol quantity need to be derived by a retrieval process or a complex observation operator.

2.12: Measurements "close to the eruption plume", this formulation might be misleading, since the article focuses on distal ash clouds.

2.16: Lidars do not provide vertical integrated column data.

2.19: It remains unclear how research flight operators know where to actually obtain the "most relevant" volcanic ash concentration.

2.25: It should be formulated what kind of estimates are highly influenced by "plume height and mass eruption rate".

4.2: "for this type of measurements in well calibrated cases." To me it is unclear what "well calibrated cases" are and whether you have those during the performed flight.

4. 29 (and Chapter 2.3 in general): To me it is not clear how the forecast error covariance matrix looks and especially how the localization is performed here.

5.12: You describe that there are no ash emissions prior to 9:00 UTC 14 April 2010, but it remains undiscussed how the volcanic ash is released during the first guess forecast

(PH, MER, VMD).

5.17: The "plume height detection data" has to be described more precisely.

5.30: "... approximates the measurements with a high accuracy" -> If I did not see the corresponding figure, I would expect the analysis to be identically equal to the observations.

6.21: "... continental Europe is all simulated in a low concentration level ..." -> This sentence needs some reformulation, since the analysis does not apply to all continental Europe, e.g. in Northern Netherlands there still appear high ash concentration values.

6.23: "Thus in a fairly large domain, the state change at measurement location also influences state variables in the surrounding areas." -> To my understanding this is caused by the chosen influence radius, which is not discussed in this study. Further, the downwind direction includes influenced states due to the transport of earlier corrected ash concentrations, especially regarding forecasts later than the assimilated time steps.

7.4: "Fig. 4 shows the difference of a number of ..." -> There is no difference plot in Fig. 4. Please be more precise, e.g. "the comparison of Fig. 4a and 4b...". And it remains unclear which "number" you refer here. Aren't these just two forecasts?

7.6: "initiated with Fig. 3c" -> Fig. 3c includes results of an assimilation process as well. Later, the forecasts using these initial values are always designated to be "without assimilation". Please resolve this issue more clearly.

7.11: It is unclear where the area of "downwind direction" is, since the wind field is nowhere described or visualized.

7.18 - 21: "... at times between 13:20 and 14:00 ... are far away from the measurement track" Actually these observations are not in the area, which is influenced by the assimilation (see Fig. 3e/3f). Therefore, it is unclear how these observations can be utilized for validation. For validation independent data has to be chosen, but it must be

selected carefully with respect to the influenced area. Additionally, these observations are in a much better agreement with the measurements than the observations taken before 13:20 UTC and after 14:00 UTC. This could lead to the interpretation that the first guess (without any assimilation) corresponds much better with the observations than the forecasts with assimilation. Please reconsider the choice of observations for validation.

7.23: Declaring the forecast results with assimilation to be "much closer to the measurements than the forecast without assimilation" for the time steps between 12:30 UTC and 13:20 UTC might be too optimistic.

Comment on Chap. 4 and 5: It remains unclear to me how the length of integration time after the assimilation can influence the forecast results. It should be discussed how the area influenced by the assimilation temporally propagates.

7.32: It has to be clarified if Antwerp and Brussels are even in the area, which is influenced by the assimilation.

8.1: In my opinion there are no continental passenger flights that operate in an altitude of 3 km. 3 km might be of special interest regarding take off and landing.

Comment on the aviation advice: Clarify that you give the aviation advice only on the strength of the results in 3 km height. Generally all model levels must be analyzed for real cases. And it should be pointed out for which exact area and which time frame the aviation advice is given.

8.21: "most of the flights in East direction" I think it is not a matter of the flight direction; it affects all flights in the ash penetrated area.

8.24: "the whole of Europe" This has to be reconsidered. Not all Europe is analyzed.

8.32: "... the assimilation impact will last at least one day." Such conclusions have to be discussed carefully. This conclusion is only valid for the areas, that are influenced by the assimilation, which changes with time, and as long as there are no high concentrated plumes transported into the area of interest. You might have to reconsider this statement especially for regions upwind to the assimilated observations.

8.34: "Only small differences are detected in the Northern part of Italy." In general I recognize only small differences at all cloud edges.

9.10: "using only one or two measurements can not produce accurate results." How do you know that at the number of measurements you used for your analysis is sufficient? Using aircraft measurements the number of observations only influences the area which can be analyzed. This does not mean, that the assimilation of one or two measurements is not valuable for a certain region.

9.25: "aviation advice can significantly benefit from the ensemble-based data assimilation process" Here it should be pointed out again, that this is only true within the influence radius/the assimilation influenced area.

9.27: "we suggest the frequency of the measurement campaign to be once per day." This cannot be said without loss of generality. Be attentive with the area that is influenced by the assimilation and the temporal change due to wind induced transport.

Comments on Figures:

Fig. 1b and 1c: could be left out, since there is no special meaning to the study or their meaning should be pointed out in the text.

Fig. 2 and 3: Which level/height is shown? Why is this level chosen?

Fig. 3e/3f: For me it is unclear how the differences of a-c and b-d can be a constant absolute value of 5. I would expect less difference between forecast and analysis at the edges of the influenced region. In addition, it is of special meaning if the assimilation induces reductions or increases of volcanic ash. Therefore, I suggest to define an extra color table for Figure 3e and 3f, which resolves the variation of the differences.

Fig. 4a/4b: The chosen colors of the aviation track might be misleading because they

[Figure]

are included in the color table.

Fig. 6: This figure shows a different domain compared to Fig. 1a,2,3,4. Be aware that not the whole domain was influenced by the assimilation.

Technical corrections:

I suggest writing UTC after every time statement.

2.22: "It has been shown. . ." should be corrected to "It was shown. . ."

2.24: "near to the eruption" -> Suggestion: "close to the eruption location"; "In this study" should be changed to "In that study"

2.30: "an Iceland eruption" -> Suggestion: "a volcanic eruption in Iceland"

3.16: "volcanic ash simulations" -> Suggestion: "volcanic ash dispersion simulations"

3.28: "All of the measurement flights were. . ." -> Suggestion: "The measurement aircraft was.." (corresponding to the reference of Fig. 1b).

4.16 and 4.21: "N state" -> Suggestion: "N states" and "N volcanic ash state" -> Suggestion: "N volcanic ash states"

4.29 and 5.5: define " ' " to represent the transposed of the matrix

5.17: PH is already defined on page 3 line 18 and again on page 5 line 18

5.27: "of some of the ensembles" -> Suggestion: "of selected ensemble members"

5.29: "but the overestimation vanishes by the assimilation process" -> Suggestion: "but the overestimation diminishes by the assimilation process significantly"

7.7: "a better forecast will be. . ." -> Suggestion: "a better forecast is expected . . ."

9.10: "can not" should be corrected to "cannot"

10.2: "in satellite data." -> suggestion: "in certain satellite data."

---

## Author Comment (AC2) · 30 May 2016

Dear Anonymous Referee #2,

Herewith we submit the revised manuscript. First we would like to thank you for the very useful comments and suggestions. We really appreciate the details provided by you. We have carefully considered all the concerns (especially the discussion of assimilation impacts on subsequent ash dispersion predictions) and made changes accordingly in the revised paper. Also the discussions on aviation advices are made by means of influenced areas. As suggested, some stated terms, interpretations and conclusions have been reformulated and clarified. Thanks to the comments, we believe the new version has been improved a lot compared to the previous version.

In the following we will give our answers and reactions to all of your comments. To make the changes easier to identify, we have numbered them.

Best regards,
G. Fu
on behalf of all co-authors

(The revised manuscript is in the latter part of this pdf.)

**Reply to the specific comments**:

1. *2.1: It should be clarified why the influence of ESPs weakens with the distance to the volcano.*

   Response: We added an explanation in lines 1.23–2.1:
   " This is mainly because (1) compared to ESPs, the plume transport becomes more and more dominant as the distance to the volcano increases (Macedonio et al., 2016); (2) the small errors in ESPs can accumulate into large errors in predicted ash concentrations after a large transport distance (Webster et al., 2012). "

2. *2.9: The difficulty of satellite data assimilation should be discussed carefully. Satellite lidar instruments like CALIOP or CATS provide highly resolved aerosol profiles. The challenge of satellite data assimilation is that the observations are often no direct measurements of the quantity of interest, but optical property measurements. Therefore the aerosol quantity need to be derived by a retrieval process or a complex observation operator.*

   Response: A discussion has now been added in lines 2.11–2.15:
   " This is because satellite observations are often not direct measurements of the quantity of interest, but optical property measurements. Therefore the aerosol quantity needs to be derived by a retrieval process or a complex observation operator. Moreover, satellite data are often two-dimensional (2D), thus are lack of sufficient vertical resolution (Bocquet et al., 2015). Note that, some satellites can provide very detailed vertical information on plumes (e.g., Cloud-Aerosol Lidar with Orthogonal Polarization (CALIOP) lidar measurements) but are spatially sparse (Winker et al., 2012). "

3. *2.12: Measurements "close to the eruption plume", this formulation might be misleading,*

*since the article focuses on distal ash clouds.*

Response: I agree. It is reformulated in lines 2.17–2.18:
" These aircraft-based measurements can be obtained in the boundaries of volcanic ash plume, which are probably the most direct volcanic ash observations possible. "

4. *2.16: Lidars do not provide vertical integrated column data.*

Response: It is corrected in lines 2.21–2.23:
" Whereas other measurements such as satellite data and LIDAR data observe optical properties that contain indirect information about concentrations and require further modeling and assumptions for getting translated into concentration estimates. "

5. *2.19: It remains unclear how research flight operators know where to actually obtain the "most relevant" volcanic ash concentration.*

Response: It is added in lines 2.24–2.27:
" Third, an aircraft is flexible in flight route to follow the ash clouds. Although an aircraft measurement plan is usually beforehand designed for a region/altitude of interest, the aircraft operator can adjust the detailed route according to the real conditions (e.g., ash concentration level, wind direction) of the plume transport in order to always obtain appropriate volcanic ash concentrations (Weber et al., 2012). "

6. *2.25: It should be formulated what kind of estimates are highly influenced by "plume height and mass eruption rate".*

Response: It is reformulated in lines 2.31–2.33:
" In that study, the focus was on the near-volcano areas where the uncertainties on plume height and mass eruption rate turned out to have a large influence on the estimates of the forecasted ash concentrations. "

7. *4.2: "for this type of measurements in well calibrated cases." To me it is unclear what "well calibrated cases" are and whether you have those during the performed flight.*

Response: Thanks for the question. The "for this type of measurements in well calibrated cases" means the aircraft data are directly calibrated with an independent reference method during the performed flight, as discussed in (Weber et al., 2010, 2012). We agree here "well calibrated cases" is a confusing term and has been reformulated in lines 4.27–4.28:
" which can be taken as the instrumental error for this type of measurements during the performed flight. "

8. *4. 29 (and Chapter 2.3 in general): To me it is not clear how the forecast error covariance matrix looks and especially how the localization is performed here.*

Response: We have added the following description in lines 6.3–6.13:
"

$$\mathbf{K}(k) = (\mathbf{f} \circ \mathbf{P}^f(k))\mathbf{H}(k)^T[\mathbf{H}(k)(\mathbf{f} \circ \mathbf{P}^f(k))\mathbf{H}(k)^T + \mathbf{R}]^{-1}, \qquad (1)$$

$$\xi_j^a(k) = \xi_j^f(k) + \mathbf{K}(k)[\mathbf{y}(k) - \mathbf{H}(k)\xi_j^f(k) + \mathbf{v}_j(k)], \qquad (2)$$

where $\mathbf{v}_j$ represents realizations of the observation error $v$.

In Eq. (1), a distance-based localization (Houtekamer and Mitchell, 1998, 2001). is obtained using a Schur product $\mathbf{f} \circ \mathbf{P}^f$ (i.e., element-wise multiplication) in order to reduce the spurious correlations caused by the finite ensemble size, which is a general problem in ensemble-based data assimilation. The correlation matrix $\mathbf{f}$ is obtained by applying a correlation function to the Euclidean distance between two points. The correlations decrease to zero beyond a certain distance. Distance-based localization can be easily implemented for eliminating the spurious correlations outside the localized region. For some applications (e.g., ozone, $CO_2$, sulfur dioxide), this approach has achieved an acceptable performance with a simple setup using a constant localization parameter of 50–100 km (Curier et al., 2012; Chatterjee et al., 2012; Barbu et al., 2009). In this study, 100 km is adopted as the localization parameter in volcanic ash assimilation of aircraft in-situ measurements. "

9. *5.12: You describe that there are no ash emissions prior to 9:00 UTC 14 April 2010, but it remains undiscussed how the volcanic ash is released during the first guess forecast (PH, MER, VMD)*

Response: We added the description in lines 6.18–6.19 which refers to lines 3.28–4.6 where a detailed description is added.

lines 6.18–6.19:
" The volcanic ash is released during the first guess forecast based on the defined ESPs (PH, MER, VMD, PSD), as discussed in Section 2.1. "
and lines 3.28–4.6:
" The input parameter PH in LOTOS-EUROS is taken from hourly based Icelandic Meteorological Office (IMO) plume height detection (Gudmundsson et al., 2012) and usually the uncertainty of PH is taken as 20 % (Bonadonna and Costa, 2013). For VMD, large explosive volcanic plumes have a typical 'umbrella' shaped vertical distribution (Sparks et al., 1997) and as such this 'umbrella' shaped VMD is adapted into LOTOS-EUROS (Fu et al., 2015; Lu et al., 2016). The PSD in LOTOS-EUROS is defined as the ESP type S2 (see definitions in Mastin et al. (2009)), in which the mass fraction of erupted debris finer than 63 $\mu m$ is 0.4. For the S2 type eruption, Durant and Rose (2009) provided the base for the PSD from their analysis of the 1992 Crater Peak, Mount Spurr event. Hence, Table 1 shows the ash distribution based on their analysis and is used in our experiments. Another input parameter MER is very hard to measure for an explosive onsetting volcano. Usually it is calculated from the plume height. Mastin et al. (2009) studied the relation between these parameters and concluded that an empirical relationship between plume height PH(km) and mass eruption rate MER (kg s$^{-1}$) is

$$PH = 2.00V^{0.241}, \quad and \quad \frac{V}{MER} = \frac{1.5e^3}{4.0e^6}. \qquad (3)$$

where V(m$^3$ s$^{-1}$) represents the volumetric flow rate. "

Table 1: Volcanic ash particle size distribution and ash bins property for LOTOS-EUROS model simulation.

| Bins | Particle Diameter | Percent of Mass | Average Particle Size ($\mu m$) |
|---|---|---|---|
| vash_1 | 250 to 2000 $\mu$m | 29 | 1125.00 |
| vash_2 | 63 to 250 $\mu$m | 31 | 156.50 |
| vash_3 | 30 to 63 $\mu$m | 12 | 46.50 |
| vash_4 | 10 to 30 $\mu$m | 18 | 20.00 |
| vash_5 | 2.5 to 10 $\mu$m | 8 | 6.25 |
| vash_6 | 0.0 to 2.5 $\mu$m | 2 | 1.25 |

10. *5.17: The "plume height detection data" has to be described more precisely.*

    Response: A description is added in lines 3.28–3.29:
    " The input parameter PH in LOTOS-EUROS is taken from the hourly based plume height detection by Icelandic Meteorological Office (IMO) (Gudmundsson et al., 2012) and usually the uncertainty of PH is taken as 20 % (Bonadonna and Costa, 2013). "

11. *5.30: "... approximates the measurements with a high accuracy" –> If I did not see the corresponding figure, I would expect the analysis to be identically equal to the observations.*

    Response: A note is added in lines 7.17–7.18:
    " Note that, the high accuracy here doesn't mean "identically equal", but "very close". "

12. *6.21: ". . . continental Europe is all simulated in a low concentration level . . ." –> This sentence needs some reformulation, since the analysis does not apply to all continental Europe, e.g. in Northern Netherlands there still appear high ash concentration values.*

    Response: It is now reformulated in lines 8.7–8.9:
    " After the assimilation process, the calculated volcanic ash concentrations in Germany, Luxembourg and the Netherlands (except in Northern Netherlands) have a lower concentration level (lower than 3000 $\mu$g m$^{-3}$) and the changes on volcanic ash state can be seen across a wide area. "

13. *6.23: "Thus in a fairly large domain, the state change at measurement location also influences state variables in the surrounding areas." –> To my understanding this is caused by the chosen influence radius, which is not discussed in this study. Further, the downwind direction includes influenced states due to the transport of earlier corrected ash concentrations, especially regarding forecasts later than the assimilated time steps.*

    Response: As suggested, the discussion is added in lines 8.12–8.15:
    " This is caused by the chosen localization radius (see Section 2.3) in the assimilation process. Further, the downwind direction includes influenced state variables due to the transport of earlier corrected ash concentrations, especially regarding forecasts later than

the assimilated time steps. ”

14. *7.4: "Fig. 4 shows the difference of a number of . . ." –> There is no difference plot in Fig. 4. Please be more precise, e.g. "the comparison of Fig. 4a and 4b. . .". And it remains unclear which "number" you refer here. Aren't these just two forecasts?*

   Response: You are right. It is corrected in line 8.26:
   " Fig. 4**a** and Fig. 4**b** show the comparison of the forecasted volcanic ash plumes with and without assimilation. ”

15. *7.6: "initiated with Fig. 3c" –> Fig. 3c includes results of an assimilation process as well. Later, the forecasts using these initial values are always designated to be "without assimilation". Please resolve this issue more clearly.*

   Response: Thanks a lot for the careful check. This is a text error which should be "initiated with Fig. 3b". It is corrected in line 8.28:
   " initiated with Fig. 3**b** and Fig. 3**d**. ”

16. *7.11: It is unclear where the area of "downwind direction" is, since the wind field is nowhere described or visualized.*

   Response: A description has now been added in lines 8.14–8.15:
   " Note that after a careful check on the wind field around the aircraft route, the term "downwind" direction means the direction of "South-East", which will be used in the following discussions. ”

17. *7.18 - 21: ". . . at times between 13:20 and 14:00 . . . are far away from the measurement track" Actually these observations are not in the area, which is influenced by the assimilation (see Fig. 3e/3f). Therefore, it is unclear how these observations can be utilized for validation. For validation independent data has to be chosen, but it must be selected carefully with respect to the influenced area. Additionally, these observations are in a much better agreement with the measurements than the observations taken before 13:20 UTC and after 14:00 UTC. This could lead to the interpretation that the first guess (without any assimilation) corresponds much better with the observations than the forecasts with assimilation. Please reconsider the choice of observations for validation.*

   Response: We agree with you. The choice of observations is reconsidered to be from 14:10 UTC to 15:00 UTC, by a careful selection of validation locations.

   The corresponding figure is re-ploted as the following figure.

   The description is reformulated in lines 9.11–9.15:
   " Furthermore, we can also see that at each validation location, the forecast with assimilation is closer to the measurements than the forecast without assimilation, and also that the overestimation is significantly reduced using assimilation. This shows that the forecast at these locations with assimilation is more accurate than the forecast without assimilation, therefore the assimilated volcanic ash state (Fig. 3**d**) is a more accurate approximation to

the real state of distal volcanic ash plume. "

18. *7.23: Declaring the forecast results with assimilation to be "much closer to the measurements than the forecast without assimilation" for the time steps between 12:30 UTC and 13:20 UTC might be too optimistic.*

Response: We agree. It is reformulated in line 9.12:
" at each validation location, the forecast with assimilation is closer to the measurements than the forecast without assimilation "

19. *Comment on Chap. 4 and 5: It remains unclear to me how the length of integration time after the assimilation can influence the forecast results. It should be discussed how the area influenced by the assimilation temporally propagates.*

Response: According to the suggestion, it is now discussed in lines 8.28–8.30:
" After the assimilation process, the assimilation influenced region temporally propagates to the downwind direction due to the meteorological drive (wind speed and direction). Thus the forecasted downwind ash concentrations are influenced along the length of integration time after assimilation. "

20. *7.32: It has to be clarified if Antwerp and Brussels are even in the area, which is influenced by the assimilation.*

    Response: After a careful check, Antwerp and Brussels are on the boundaries of the influenced area. For avoiding misleading, we removed two cities since the influence on them is not obvious. The new description is in lines 9.20–9.21:
    " For this investigation, 5 big cities around the measurement route are selected (see Fig. 5**a**). They are Dortmund, Cologne, Luxembourg in the downwind direction and Amsterdam, Rotterdam in the upwind direction. "
    The corresponding figure is changed to:

[Figure]

**a**

| Germany (DEU) | 1 | Dortmund |
|---|---|---|
| | 2 | Cologne |
| Luxembourg (LUX) | 3 | Luxembourg |
| Netherlands (NLD) | 4 | Amsterdam |
| | 5 | Rotterdam |

**b**

| City Number / Simulation | | downwind | | | upwind | |
|---|---|---|---|---|---|---|
| | | 1 | 2 | 3 | 4 | 5 |
| PM$_{10}$ | Without assimilation($\mu$g m$^{-3}$) | 574.0 | 817.5 | 568.8 | 1055.3 | 863.4 |
| | With assimilation($\mu$g m$^{-3}$) | 112.1 ↓ | 219.1 ↓ | 262.5 ↓ | 717.8 ↓ | 724.2 ↓ |
| | Impact rate | 80.5% | 73.2% | 53.9% | 32.0% | 16.1% |
| PM$_{2.5}$ | Without assimilation($\mu$g m$^{-3}$) | 243.7 | 344.6 | 216.1 | 387.9 | 306.3 |
| | With assimilation($\mu$g m$^{-3}$) | 49.3 ↓ | 90.5 ↓ | 97.3 ↓ | 259.3 ↓ | 256.6 ↓ |
| | Impact rate | 79.8% | 73.7% | 55.0% | 33.2% | 16.2% |

21. *8.1: In my opinion there are no continental passenger flights that operate in an altitude of 3 km. 3 km might be of special interest regarding take off and landing.*

    Response: It is corrected in lines 9.21–9.23:
    " The evaluation height is chosen at 3 km. For some continental or intercontinental passenger flights, 3 km might be of special interest regarding taking off and landing. "

22. *Comment on the aviation advice: Clarify that you give the aviation advice only on the strength of the results in 3 km height. Generally all model levels must be analyzed for real cases. And it should be pointed out for which exact area and which time frame the aviation*

*advice is given.*

Response: It is now reformulated in lines 10.13–10.15:
" Note that we give the aviation advice only on the strength of the results at 3 km height. Generally all model levels must be analyzed for real cases. And the real aviation advice also includes for which exact area and which time frame the advice is given. "

23. *8.21: "most of the flights in East direction" I think it is not a matter of the flight direction; it affects all flights in the ash penetrated area.*

Response: It is corrected in lines 10.9–10.10:
" imposing such a no-fly zone will affect all flights in the ash penetrated area and subsequently leads to a huge economic loss. "

24. *8.24: "the whole of Europe" This has to be reconsidered. Not all Europe is analyzed.*

Response: It is reformulated in lines 10.11–10.12:
" The sky in large parts of Europe is open for commercial flights, because except in small parts of the Netherlands ash concentrations all over the domain of interest are lower than 3000 $\mu$g m$^{-3}$. "

25. *8.32: ". . . the assimilation impact will last at least one day." Such conclusions have to be discussed carefully. This conclusion is only valid for the areas, that are influenced by the assimilation, which changes with time, and as long as there are no high concen- trated plumes transported into the area of interest. You might have to reconsider this statement especially for regions upwind to the assimilated observations.*

Response: We agree. The statement has been reconsidered in lines 10.20–10.25:
" Since there are clear differences between the two cases, the assimilation impact can last one day. Note that this impact duration is only valid for the areas (especially for regions downwind to the assimilated observations) that are influenced by the assimilation, which changes with time. When forecasting 24 hours (Fig. 6**c**), differences still can be observed, but the impact of assimilation is obviously getting much smaller (compared to Fig. 6**a** and Fig. 6**b**). Small differences are visible at cloud boundaries. Actually we also examined the assimilation impact in the forecast of the next day and observed only very small differences. "

26. *8.34: "Only small differences are detected in the Northern part of Italy." In general I recognize only small differences at all cloud edges.*

Response: It is corrected in line 10.24:
" Small differences are visible at cloud boundaries. "

27. *9.10: "using only one or two measurements can not produce accurate results." How do you know that at the number of measurements you used for your analysis is sufficient?*

*Using aircraft measurements the number of observations only influences the area which can be analyzed. This does not mean, that the assimilation of one or two measurements is not valuable for a certain region.*

Response: We agree that it is incorrect to state "using only one or two measurements can not produce accurate results.". The sentence is changed in line 11.1:
" It was shown that all the assimilation steps contribute to the final result. "

28. *9.25: "aviation advice can significantly benefit from the ensemble-based data assim- ilation process" Here it should be pointed out again, that this is only true within the influence radius/the assimilation influenced area.*

    Response: As suggested, it has been added in line 11.15–11.17:
    " Investigation shows that the accuracy of aviation advice within the assimilation influenced area can significantly benefit from the ensemble-based data assimilation process. The computer experiment revealed that the time period of the improvement effect on the areas downwind to the assimilated observations can be taken as 24 hours. "

29. *9.27: "we suggest the frequency of the measurement campaign to be once per day." This cannot be said without loss of generality. Be attentive with the area that is influenced by the assimilation and the temporal change due to wind induced transport.*

    Response: It is reformulated in lines 11.17–11.20:
    " we suggest to schedule an aircraft measurement campaign at a frequency of once per day. This can be used to provide guidelines for planning future regional measurement tasks. The suggested frequency should be adjusted by the temporal strength (due to wind induced transport) on the assimilation influenced area. "

**Reply to Comments on Figures**:

1. *Fig. 1b and 1c: could be left out, since there is no special meaning to the study or their meaning should be pointed out in the text.*

    Response: It has been changed as suggested, see the revised manuscript (latter part of this pdf).

2. *Fig. 2 and 3: Which level/height is shown? Why is this level chosen?*

    Response: The height information of "3 km" is added in Fig. 2 and 3 (see the revised manuscript). The evaluation height is chosen to be 3 km because the measurements are taken at altitudes around this height, which is described in lines 4.31–4.32 (" The measurements took place at heights around 3 km (Weber et al., 2012) ").

3. *Fig. 3e/3f: For me it is unclear how the differences of a-c and b-d can be a constant absolute value of 5. I would expect less difference between forecast and analysis at the edges*

*of the influenced region. In addition, it is of special meaning if the assimilation induces reductions or increases of volcanic ash. Therefore, I suggest to define an extra color table for Figure 3e and 3f, which resolves the variation of the differences.*

Response: As suggested, we re-plotted Figure 3e and 3f.

[Figure]

4. *Fig. 4a/4b: The chosen colors of the aviation track might be misleading because they are included in the color table.*

Response: The color for aviation track is changed to black.

5. *Fig. 6: This figure shows a different domain compared to Fig. 1a,2,3,4. Be aware that not the whole domain was influenced by the assimilation.*

   Response: Thank you for the careful check. It is explained in the caption of Fig. 6: " A larger domain is chosen in this figure (compared to Fig. 2,3,4) to demonstrate the downwind change (with time) of the assimilation influenced area. "

**Reply to Technical corrections**:

1. As suggested, we write UTC after every time statement.

2. "It has been shown. . ." is corrected to "It was shown. . ." in line 2.29.

3. "near to the eruption" is changed to "close to the eruption location" in line 2.31; "In this study" is changed to "in that study" in line 2.31.

4. "an Iceland eruption" is changed to "a volcanic eruption in Iceland" in lines 3.2–3.3.

5. "volcanic ash simulations" is changed to "volcanic ash dispersion simulations" in line 3.24.

6. "All of the measurement flights were. . ." is changed to " The measurement aircraft was..." in line 4.23.

7. "N state" is changed to "N states" in line 5.11; "N volcanic ash state" is changed to "N volcanic ash states" in line 5.16.

8. As suggested, the "transpose representation" is added in line 5.25: " where the superscript "$T$" represents the transpose of the matrix. "

9. The repeat of PH definition in Section 3.1 is deleted, which is only in Section 2.1.

10. "of some of the ensembles" is changed to "of selected ensemble members" in line 7.14.

11. "but the overestimation vanishes by the assimilation process" is changed to " but the overestimation diminishes by the assimilation process significantly" in line 7.16.

12. "a better forecast will be. . ." is changed to " a better forecast is expected . . " in line 8.31.

13. "can not" is changed to " cannot" in line 11.26.

14. "in satellite data" is changed to " in certain satellite data" in line 11.27.

The revised manuscript starts from next page.

[revised manuscript text omitted]